# LACONIC: LENGTH-AWARE CONSTRAINED REINFORCEMENT LEARNING FOR LLM

## ABSTRACT

Reinforcement learning (RL) has enhanced the capabilities of large language models (LLMs) by enabling self-evolution through reward-driven training. Nevertheless, this process can introduce excessively long responses that inflate inference latency and computational overhead. To address this issue, existing RL-based length control methods often incorporate fixed penalties or heuristic reward shaping to encourage outputs of a desired length. However, such strategies may misalign the optimization objective with the underlying task, resulting in suboptimal performance and limited generalization across model architectures and datasets. In this work, we propose LACONIC, a lightweight reinforcement learning method that enforces a target token budget during training. Specifically, we update policy models using an augmented objective that combines the task reward with a length-based cost applied only to tokens exceeding the specified budget. Furthermore, to balance brevity and task performance, the cost scale is adjusted online throughout training. This formulation directly optimizes task reward subject to an explicit token budget constraint, delivering precise and performance-preserving length control. Across mathematical reasoning models and datasets, LACONIC preserves or improves pass@1 while reducing output length by up to 43%. It maintains out-of-domain performance on general knowledge and multilingual benchmarks with a 44% reduction in tokens. Moreover, LACONIC integrates into standard RL fine-tuning with no inference changes and minimal deployment overhead.

## 1 INTRODUCTION

Large language models (LLMs) such as GPT [OpenAI et al., 2024; OpenAI, 2025], Gemini [Comanici et al., 2025], DeepSeek [DeepSeek-AI, 2025], and Claude[Anthropic, 2025] have witnessed unprecedented success in its applications from software agents to enterprise analytics [Team et al., 2025; Li et al., 2025b; Jin et al., 2025; Feng et al., 2025]. The impressive capabilities of LLMs have been significantly enhanced by reinforcement learning based fine-tuning [Li et al., 2025a; Wu et al., 2025; Li et al., 2025b], a procedure that aligns pretrained models with task-specific rewards through interaction with an environment. This process has been pivotal in refining LLM reasoning skills, enhancing generalization, and achieving state-of-the-art performance across diverse benchmarks [Wang et al., 2024; Hsiao et al., 2025; Shi et al., 2025; Qu et al., 2025]. However, RL-tuned language models often suffer from generating unnecessarily long thinking traces. This problem is particularly acute on reasoning and mathematics tasks, where the model is asked to spell out logical steps [Chen et al., 2025; Sui et al., 2025]. In practice, excessive verbosity inflates training and inference time, increases memory pressure, and ultimately degrades user experience.

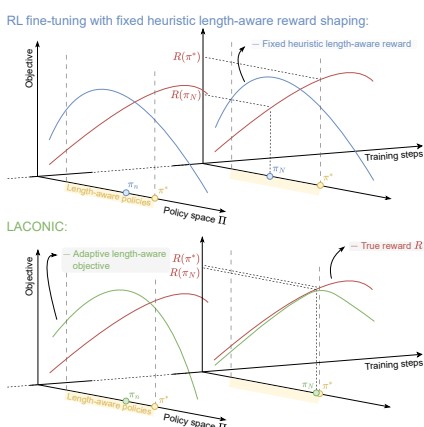

Figure 1: Heuristic reward shaping yields suboptimal task rewards, while adaptive length-aware objective realigns optimization with true rewards, preserving performance under length control.

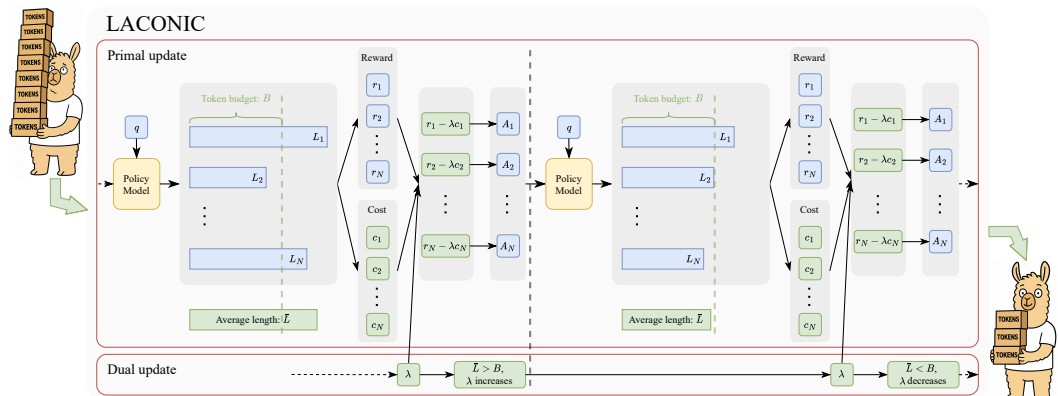

Figure 2: Illustration of LACONIC. LACONIC alternates two steps: (1) in a primal update, the policy model is updated on an augmented objective that trades off task reward $r$ with a length-aware cost $c$ scaled by the dual variable $\lambda$; (2) in a dual update, $\lambda$ is adaptively updated to enforce a token budget constraint $B$ by increasing when the average length $\bar{L}$ exceeds the budget $B$ and decreasing otherwise. Together, these updates maximize task reward while meeting the budget on average.

Recent work on length-aware LLMs has explored positional encoding, prompt engineering, and post-generation truncation [Li et al., 2025a; Wu et al., 2025; Li et al., 2025b]. A straightforward method is to design new reward functions to incorporate response length signals into RL-tuning [Aggarwal & Welleck, 2025; Cheng et al., 2025; Huang et al., 2025; Yuan et al., 2025]. These methods hard-code a length-aware reward, typically with a fixed penalty or heuristic shaping that stays fixed throughout training. Fine-tuning with these rewards optimizes a surrogate objective that is misaligned with true task reward, and often demands per-setting hyperparameter tuning. Figure 1 (top) visualizes this issue by showing a sketch of training process. The fixed heuristic objective differs from the true objective, so optimizing it yields policies with suboptimal task rewards.

In this paper, we address length control in RL-tuning by maximizing task reward subject to an average token budget constraint. We introduce LACONIC (**L**ength-**A**ware **Con**strained Pol**ic**y Optimization), a primal-dual algorithm. During training, the model samples candidate responses for each prompt. Besides the task reward (e.g., correctness or usefulness) as in the standard RL-tuning, we assign to each candidate response also a length-aware cost proportional to its budget violation. We then construct a learning signal that combines task reward with length cost, scaled by an adaptively learned multiplier. We alternatively update the policy model and the multiplier. The policy is updated by a policy optimization step where advantages and objectives are calculated from the constructed signal. Then we update the multiplier based on the average response length of the current batch. We raise the multiplier if the current batch violates the token budget constraint on average, and lower it if the current batch falls short. This feedback automatically steers the model's average output length towards the token budget. Particularly, when the model consistently stays within the token budget, the multiplier naturally drops to zero and our training steps reduce to standard RL-tuning steps, allowing the model to recover task rewards with shortened responses. As illustrated in figure 1, LACONIC adopts an adaptive objective that dynamically align the optimization with true rewards, steering the policy towards the length-aware optimum.

We conduct extensive experiments to evaluate our method LACONIC and present the evaluation results in section 3. We apply LACONIC to fine-tune two reasoning models DeepScaleR-1.5B-Preview [Luo et al., 2025] and Qwen2.5-Math-1.5B-Instruct [Yang et al., 2024]. The experimental results show that our LACONIC-tuned models can significantly outperform existing length control baseline L1 [Aggarwal & Welleck, 2025] and match GRPO on pass@1 across common mathematics benchmarks, while effectively shortening response lengths by using 44% and 26% fewer tokens than the base model and L1-Max respectively. LACONIC also preserves accuracy on benchmarks outside our RL-tuning domain while reducing response length by 44% compared to GRPO. Furthermore, we perform ablation analysis on the hyperparameters of our method in section 4. Ablation experiments show that LACONIC provides robust and length control.

## 2 METHODOLOGY

### 2.1 PRELIMINARY BACKGROUND

RL fine-tuning casts text generation as a sequential decision process, where the prompt $q$ together with the partial output sequence constitutes the state, selecting the next token is the action, and the language model parameterized by $\theta$ serves as the policy $\pi_\theta$ that maps states to action probabilities. After generating the response, the model receives task rewards $r(q, o)$ assigned by a reward model. Policy gradient algorithms such as Proximal Policy Optimization (PPO) [Schulman et al., 2017] and Group Relative Policy Optimization (GRPO) [Shao et al., 2024a] then translate such rewards into token-level gradients, so that the model can be updated to increase the corpus-level expected rewards, i.e, $\max_\theta \mathbb{E}_{q \sim P(Q), o \sim \pi_\theta(\cdot|q)}[r(q, o)]$. In practice, GRPO updates the policy model's parameters $\theta$ by optimizing the following surrogate objective

$$\mathcal{J}(\theta) = \mathbb{E}_{q, \{o_i\}_{i=1}^G} \left[ \frac{1}{G} \sum_{i=1}^{G} \frac{1}{|o_i|} \sum_{t=1}^{|o_i|} \min(\rho_{i,t} A_{i,t}, \text{clip}(\rho_{i,t}, 1-\varepsilon, 1+\varepsilon) A_{i,t}) - \beta \mathbb{D}_{\text{KL}}[\pi_\theta \| \pi_{\text{ref}}] \right], \quad (1)$$

where $\rho_{i,t} = \frac{\pi_\theta(o_{i,t}|q, o_{i,<t})}{\pi_{\theta_{\text{old}}}(o_{i,t}|q, o_{i,<t})}$ is the likelihood ratio, and $A_{i,t}$ is the group-relative advantage. The likelihood ratio clipping and an extra KL-divergence penalty are adopted to stabilize policy updates.

### 2.2 LACONIC: LENGTH-AWARE CONSTRAINED POLICY OPTIMIZATION

To add explicit control over response length, we extend standard RL fine-tuning to a constrained setting that maximizes task reward under an average token count constraint $B$, a pre-specified budget reflecting deployment targets such as latency and computational resources in practice. We treat the token budget as a given hyperparameter and learn a policy whose average response length respects the budget $B$. Formally, letting $L(o)$ denote the length of response $o$, we address

$$\max_\theta \mathbb{E}_{q \sim P(Q), o \sim \pi_\theta(\cdot|q)}[r(q, o)], \text{ s.t. } \mathbb{E}_{q \sim P(Q), o \sim \pi_\theta(\cdot|q)}[L(o)] \leq B. \quad (2)$$

In this formulation, we enforce a corpus-level average token budget rather than a more strict per-sequence length constraint. Response lengths naturally vary across prompts. As an example, a math olympiad problem typically requires more tokens to solve than a simple arithmetic query. The average budget constraint in equation (2) allows the model to generate more tokens where they improve reward and shorten responses on easier cases while still meeting the overall budget constraint.

**Length-aware cost.** In order to enforce the budget constraint during training, we assign each response a sequence-level cost that measures its budget violation. Specifically, for a prompt $q$ and a response $o$ generated by the policy model $\pi_\theta$, we define the cost as

$$c(q, o) = \max\left\{ \frac{L(o) - B}{B}, 0 \right\}. \quad (3)$$

The cost is zero for responses no longer than $B$, and increases linearly with the number of over-budget tokens. This design discourages unnecessary verbosity while preserving exploration as the model can freely use up to $B$ tokens without penalty. While task rewards often lie in $[0, 1]$, response lengths can range widely from a few hundreds to over 100K across backbones, prompt distributions, and output length caps in practice. We add the $1/B$ normalization to keep the cost on a comparable scale across different magnitudes of lengths and budgets.

**Lagrangian reward.** We incorporate the cost into the learning objective via a Lagrangian relaxation. With a nonnegative multiplier $\lambda \geq 0$, also known as a dual variable, we define the Lagrangian reward

$$\ell_\lambda(q, o) = r(q, o) - \lambda \cdot c(q, o). \quad (4)$$

**Primal updates.** To tackle the constrained optimization in equation (2), we alternate a primal (policy) update and a dual (multiplier) update. In the primal update, we hold $\lambda$ fixed and take a policy gradient step to optimize expected Lagrangian reward, i.e., $\max_\theta \mathbb{E}_{q,o}[\ell_\lambda(q, o)]$. This is essentially the standard RL-tuning objective with the task reward $r$ replaced by the Lagrangian reward $\ell_\lambda$. Therefore,

---

**Algorithm 1:** LACONIC (Length-Aware Constrained Policy Optimization)

---

**Input:** initial policy model $\pi_{\theta_{\text{init}}}$; reward models $r_\varphi$; task prompts $\mathcal{D}$; token budget $B$; step size $\eta$;
        initial dual variable $\lambda_{\text{init}}$

1   policy model $\pi_\theta \leftarrow \pi_{\theta_{\text{init}}}$
2   dual variable $\lambda \leftarrow \lambda_{\text{init}}$
3   **for** *iteration* $= 1, \ldots, I$ **do**
4     reference model $\pi_{\text{ref}} \leftarrow \pi_\theta$
5     **for** *step* $= 1, \ldots, M$ **do**
6       Sample a batch $\mathcal{D}_b$ from $\mathcal{D}$
7       Update the old policy model $\pi_{\theta_{\text{old}}} \leftarrow \pi_\theta$
8       Sample $G$ outputs $\{o_i\}_{i=1}^G \sim \pi_{\theta_{\text{old}}}(\cdot|q)$ for each question $q \in \mathcal{D}_b$
9       Compute rewards $\{r_i\}_{i=1}^G$ for each sampled output $o_i$ by running $r_\varphi$
10      Compute costs $\{c_i\}_{i=1}^G$ for each sampled output $o_i$ by equation (3)
11      Compute Lagrangian rewards $\{\ell_{\lambda,i}\}_{i=1}^G$ for each sample output by equation (4)
12      Compute advantages $A_{i,t}$ for the $t$-th token of $o_i$ by equation (5)
        // Primal update
13      Update the policy model $\pi_\theta$ by maximizing the GRPO-style objective
        // Dual update
14      Update the dual variable $\lambda$ by equation (6)
15     Update $r_\varphi$
**Output:** $\pi_\theta$

---

we adapt the GRPO-style policy update with the objective computed from $\ell_\lambda$. Specifically, for each prompt $q$, we sample a group of candidate outputs $\mathbf{o} = \{o_1, o_2, \ldots, o_G\}$ from the current policy model $\pi_\theta$, and compute their task rewards $\mathbf{r} = \{r_1, r_2, \ldots, r_G\}$ and costs $\mathbf{c} = \{c_1, c_2, \ldots, c_G\}$ by equation (3). We then compute the Lagrangian rewards $\boldsymbol{\ell}_\lambda = \{\ell_{\lambda,1}, \ell_{\lambda,2}, \ldots, \ell_{\lambda,G}\}$ where $\ell_{\lambda,i} = r_i - \lambda c_i$. For each token $o_{i,t}$, we construct the GRPO-style advantage as the normalized Lagrangian reward, i.e.,

$$A_{i,t} = \widetilde{\ell}_{\lambda,i} = \frac{\ell_{\lambda,i} - \text{mean}(\boldsymbol{\ell}_\lambda)}{\text{std}(\boldsymbol{\ell}_\lambda)}. \tag{5}$$

The policy model is optimized by maximizing the GRPO objective in equation (1) where advantages are calculated by equation (5).

**Dual updates.** In the dual update, we adjust the dual variable $\lambda$ based on how the batch compares to the token budget. Let $\bar{L}$ denote the average response length of the current batch. We update

$$\lambda \leftarrow \max\left\{\lambda + \eta\left(\frac{\bar{L}}{B} - 1\right), 0\right\}, \tag{6}$$

with the step size $\eta > 0$. When the batch violates the token budget on average ($\bar{L} > B$), the update increases $\lambda$, raising the effective price of tokens in $\ell_\lambda$. Longer responses then receive lower (often negative) advantages than shorter responses with similar task rewards, so the next primal update shifts the policy toward shorter outputs. If the batch falls within the budget, $\lambda$ relaxes towards 0. Notably, when $\lambda = 0$, $\ell_\lambda$ reduces to the task reward $r$, and the next primal update is exactly a GRPO step. This feedback adapts $\lambda$ to track the budget constraint throughout training as the policy and length distribution evolve.

We present LACONIC in algorithm 1 and illustrate the workflow with two sample steps in figure 2.

## 3 EXPERIMENT

### 3.1 EXPERIMENTAL SETUP

**Models and Datasets.** For the training dataset, we use DeepScaleR-Preview-Dataset [Luo et al., 2025], a math dataset containing 40.3K rows of question-answer pairs sampled from AIME (prior to 2023), AMC (prior to 2023), Omni-MATH [Gao et al., 2024], and STILL [Min et al., 2024]. For

Table 1: Evaluation results across five math benchmarks.

| Model | AIME2024 | | MATH | | Minerva | | Olympiad | | Macro Average | |
|---|---|---|---|---|---|---|---|---|---|---|
| | Pass@1 | # Tokens | Pass@1 | # Tokens | Pass@1 | # Tokens | Pass@1 | # Tokens | Pass@1 | # Tokens |
| DeepScaleR-1.5B | | | | | | | | | | |
| + GRPO | 26.25 | 4462 | 83.44 | 1657 | 28.81 | 1982 | **46.59** | 2600 | 46.27 | 2675 |
| + L1-Exact | 21.88 | 3754 | 82.44 | 3614 | 28.70 | 3612 | 44.55 | 3687 | 44.39 ↓1.88 | 3667 ↑37% |
| + L1-Max | 25.00 | 2879 | 83.50 | 1794 | 28.28 | 1624 | 44.96 | 2025 | 45.44 ↓0.83 | 2080 ↓22% |
| **+ LACONIC** | **27.50** | **2451** | **83.72** | **1000** | **28.84** | **1113** | 45.82 | **1496** | 46.47 ↑0.2 | **1515 ↓43%** |
| Qwen-Math-1.5B | | | | | | | | | | |
| + GRPO | 11.46 | 952 | **74.86** | 570 | 25.14 | 656 | **39.84** | 808 | **37.89** | 747 |
| + L1-Exact | 9.79 | 1309 | 69.69 | 988 | 22.52 | 1277 | 37.17 | 1128 | 34.79 ↓3.1 | 1176 ↑57% |
| + L1-Max | 11.04 | 1229 | 70.39 | 779 | 22.93 | 1239 | 37.07 | 1063 | 35.36 ↓2.53 | 1078 ↑44% |
| **+ LACONIC** | **11.46** | **674** | 73.85 | **464** | **25.25** | **524** | 38.39 | **603** | 37.24 ↓0.65 | **566 ↓24%** |

base models, we use DeepScaleR-1.5B-Preview [Luo et al., 2025] (**DeepScaleR-1.5B** for short) and Qwen2.5-Math-1.5B-Instruct [Yang et al., 2024] (**Qwen-Math-1.5B** for short). DeepScaleR-1.5B is a 1.5B-parameter reasoning model fine-tuned from DeepSeek-R1-Distilled-Qwen-1.5B [DeepSeek-AI, 2025] on DeepScaleR-Preview-Dataset, and Qwen-Math-1.5B is a 1.5B-parameter instruction-tuned math model. In line with previous works, we set the maximum response length to 4K tokens per prompt during training and 8K tokens during evaluation for DeepScaleR-1.5B. We set the maximum response length to 2K tokens per prompt during training and evaluation for Qwen-Math-1.5B, because the model supports only up to 4K context lengths.

**Baselines.** We fine-tune the base models with the following algorithms to serve as baselines and compare with our algorithm LACONIC: (i) GRPO [Shao et al., 2024a]: the standard RL-tuning algorithm originally used in the post-training of DeepScaleR-1.5B and Qwen math models; (ii) L1-Exact [Aggarwal & Welleck, 2025]: a length-controlled policy optimization algorithm that fine-tunes models to satisfy exact token-length constraints; (iii) L1-Max [Aggarwal & Welleck, 2025]: a variant of L1-Exact that further fine-tunes models to observe a maximum token budget.

**Evaluation.** We evaluate models on 4 common mathematics benchmarks: AIME2024, MATH [Hendrycks et al., 2021], Minerva [Lewkowycz et al., 2022], and Olympiad-Bench [He et al., 2024]. To assess the mathematical reasoning ability of the models, we report pass@1, the fraction of questions for which the model's first response matches the correct answer. To quantify the verbosity of the model's output, we report the average response length.

## 3.2 MAIN RESULTS

In this section, we first present in table 1 the main results of all baselines and LACONIC on the 4 mathematics benchmarks. Then we present in table 2 the results on out-of-domain benchmarks, GPQA, LSAT, and MMLU, which probe general knowledge and logic reasoning.

**LACONIC outperforms existing length-control methods and matches vanilla RL-tuning while significantly reducing response lengths.** On DeepScaleR-1.5B, after fine-tuning, LACONIC outperforms GRPO and both L1 variants on pass@1 while significantly shortening its answers by emitting 43%, 58%, and 26% fewer tokens per prompt on average than GRPO, L1-Exact, and L1-Max respectively. Specifically, LACONIC-tuned model outperforms the three baselines with the highest pass@1 in AIME2024, MATH, and Minerva benchmarks while using significantly fewer tokens. On Qwen-Math-1.5B, after fine-tuning, LACONIC virtually preserves the pass@1 performance of GRPO and outperforms both L1 variants, while substantially shortening its answers by emitting 24%, 52%, and 47% fewer tokens per prompt on average than GRPO, L1-Exact, and L1-Max respectively. Notably, LACONIC generates least tokens across all benchmarks on both base models with comparable or higher pass@1.

**LACONIC preserves out-of-domain (OOD) capabilities.** LACONIC preserves GRPO's macro average accuracy while generating 44% fewer tokens on average. Our method matches vanilla RL-

Table 2: Evaluation results across out-of-domain (OOD) benchmarks.

| Model | GPQA | | LSAT | | MMLU | | Macro Average | |
|---|---|---|---|---|---|---|---|---|
| | Accuracy | # Tokens | Accuracy | # Tokens | Accuracy | # Tokens | Accuracy | # Tokens |
| DeepScaleR-1.5B | | | | | | | | |
| + GRPO | **28.88** | 2229 | 24.6 | 3426 | 43.30 | 719 | **32.26** | 2125 |
| + L1-Exact | 22.88 | 1475 | 25.19 | **1374** | 36.67 | 766 | 28.25 ↓4.01 | 1205 ↓43% |
| + L1-Max | 28.72 | 1773 | **26.58** | 2321 | 38.48 | 863 | 31.26 ↓1.00 | 1652 ↓22% |
| + **LACONIC** | 27.20 | **1167** | 24.53 | 1923 | **43.36** | **497** | 31.69 ↓0.57 | **1196** ↓44% |

tuning on MMLU and LSAT with 31% and 44% fewer tokens respectively, and trades 1.68 percentage points on GPQA for a 48% length reduction. Compared with both L1 variants, LACONIC attains higher macro pass@1 with fewer tokens. The results indicate strong OOD task reward preservation with substantially shorter outputs.

## 4 FURTHER ANALYSIS

In this section, we present additional ablation analysis related to hyperparameters, and examine the computational resources required by LACONIC, including runtime, FLOPs, and memory usage.

### 4.1 ABLATION ANALYSIS ON BUDGET $B$

We vary the token budget $B \in \{3000, 2000, 1750, 1500\}$ on DeepScaleR-1.5B while keeping all other settings and hyperparameters (including the dual step size) fixed, and train for 300 steps. Figure 3 shows the training dynamics of (a) accuracy reward; (b) average response length; (c) dual variable $\lambda$; and (d) average response length to budget ratio $\bar{L}/B$. We evaluate the step-300 checkpoints on the four mathematics benchmarks. Table 3 reports pass@1 and average response lengths.

**LACONIC provides reliable, hyperparameter-tuning-free length control.** Across token budgets, training rapidly drives the average response length under the budget and maintains it near the budget once stabilized. Even under tight constraints on a backbone that naturally produces long responses, LACONIC keeps the average length near the budget. Although prompt distributions during evaluation differ from training, table 3 shows a clear monotonic relation between the token budget $B$ and average response length. In practice, $B$ acts as a single knob and no re-tuning of other hyperparameters is required to achieve effective length control.

**LACONIC achieves better reward with less tokens than GRPO and existing length control methods across a wide range of token budgets.** We compare the evaluation results of LACONIC with token budgets ranging from 1500 to 3000 in table 3 to GRPO and both L1 variants in table 1. Across a wide range of token budgets, LACONIC consistently outperforms all baselines (GRPO and L1) on most benchmarks while using substantially fewer tokens at the same time. This shows the superiority of LACONIC to preserve reward under constrained token budgets.

Overall, these results demonstrate LACONIC's controllability (the achieved lengths drop as the budget shrinks), stability (training lengths remain near the budget), and decoupling (changing token budgets $B$ require no re-tuning of other hyperparameters).

### 4.2 ABLATION ANALYSIS ON DUAL STEP SIZE $\eta$

We vary the step size for dual updates $\eta \in \{0.002, 0.01, 0.02\}$ while keeping all other settings fixed and train Qwen-Math-1.5B for 350 steps under a token budget $B = 550$. We set the token budget $B = 550$ for all runs. For $\eta = 0.02$, we also set a ceiling $\lambda_{\max} = 0.1$ for the dual variable. Figure 4 plots the training dynamics of (a) accuracy reward; (b) average response length; (c) dual variable $\lambda$; (d) average response length to budget ratio $\bar{L}/B$. We evaluate the step-350 checkpoints on the 4 mathematics benchmarks, and report the pass@1 and average response lengths in table 4.

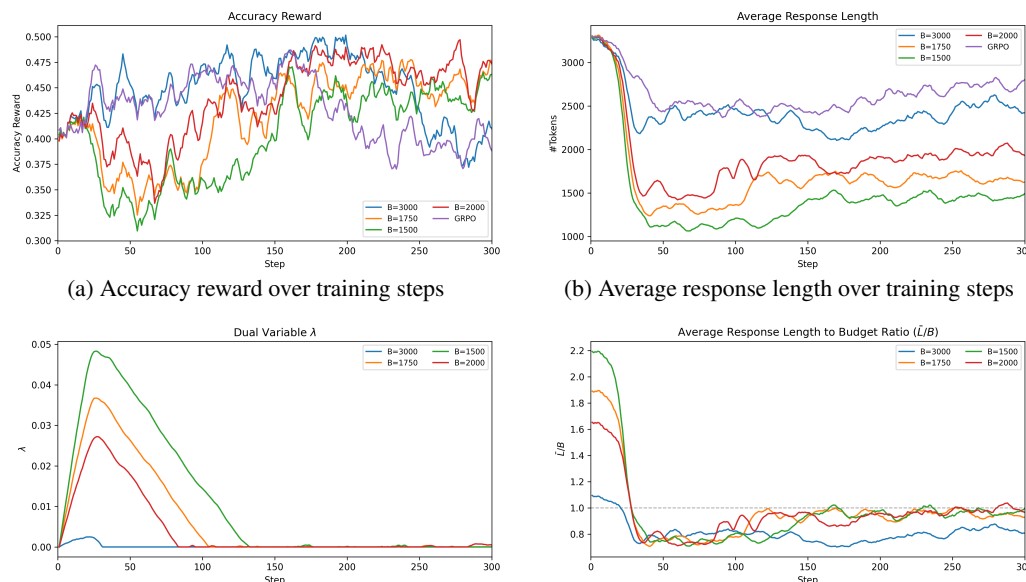

(a) Accuracy reward over training steps

(b) Average response length over training steps

(c) Dual variable $\lambda$ over training steps

(d) Average length to budget $\bar{L}/B$ over training steps

Figure 3: Ablation on token budget $B$. All plots are smoothened over 10 steps except (c) dual variable.

Table 3: Evaluation results across 4 math benchmarks.

| Model | AIME2024 | | MATH | | Minerva | | Olympiad | | Macro Average | |
|---|---|---|---|---|---|---|---|---|---|---|
| | Pass@1 | # Tokens | Pass@1 | # Tokens | Pass@1 | # Tokens | Pass@1 | # Tokens | Pass@1 | # Tokens |
| DeepScaleR-1.5B | | | | | | | | | | |
| + GRPO | 26.25 | 4462 | 83.44 | 1657 | 28.81 | 1982 | 46.59 | 2600 | 46.27 | 2675 |
| + LACONIC (3000) | 29.79 | 3642 | 84.46 | 1490 | 29.76 | 1921 | 47.07 | 2386 | 47.77 ↑1.50 | 2360 ↓12% |
| + LACONIC (2000) | 30.83 | 3558 | 85.70 | 1454 | 31.02 | 1839 | 48.94 | 2357 | 49.12 ↑2.85 | 2302 ↓14% |
| + LACONIC (1750) | 30.21 | 3477 | 84.62 | 1388 | 30.52 | 1659 | 48.78 | 2221 | 48.53 ↑2.26 | 2186 ↓18% |
| + LACONIC (1500) | 27.50 | 3124 | 84.39 | 1297 | 31.60 | 1494 | 48.65 | 2048 | 48.04 ↑1.77 | 1991 ↓26% |

**LACONIC is insensitive to the dual step size $\eta$.** Across different dual step sizes, LACONIC delivers consistent length control. With $\eta = 0.002$ versus 0.01, the training dynamics of average response length are virtually identical throughout, and the final policies match after stabilization. With $\eta = 0.02$ (plus a $\lambda$-ceiling), the curves settle into the same stabilized dynamics and reach comparable final reward. This shows that LACONIC is robust to an order-of-magnitude change in the dual step size $\eta$.

**LACONIC works consistently with a $\lambda$-ceiling.** Across runs with and without a ceiling, the training curves and final policies are essentially the same once stabilized. In both cases LACONIC tracks the token budget and recovers reward. The $\lambda$-ceiling is a preference safeguard. Our primary goal is to favor correct responses over incorrect ones, even when correct answers are longer. With the Lagrangian reward, an excessively large $\lambda$ can flip preferences and make very short but incorrect outputs score higher than correct long outputs. Specifically, for an indicator reward function $r(q, o) = \mathbb{1}_{\{o \text{ is correct}\}}$, we require $\ell_\lambda(q, o_c) = 1 - \lambda(L(o_c) - B)/B > 0$, where $o_c$ is any correct response. This translates to an upper bound of $\lambda < B/(L(o_c) - B)$ for any correct response $o_c$. It suffices to set a ceiling of $\lambda_{\max} = B/(L_{\max} - B)$, where $L_{\max}$ is the maximum response length cap. In cases where a $\lambda$-ceiling $\lambda_{\max}$ is present, we augment the dual update to a projected version: $\lambda \leftarrow \text{clip}(\lambda + \eta \left( \frac{\bar{L}}{B} - 1 \right), 0, L_{\max})$.

### 4.3 COMPUTATIONAL RESOURCE ANALYSIS

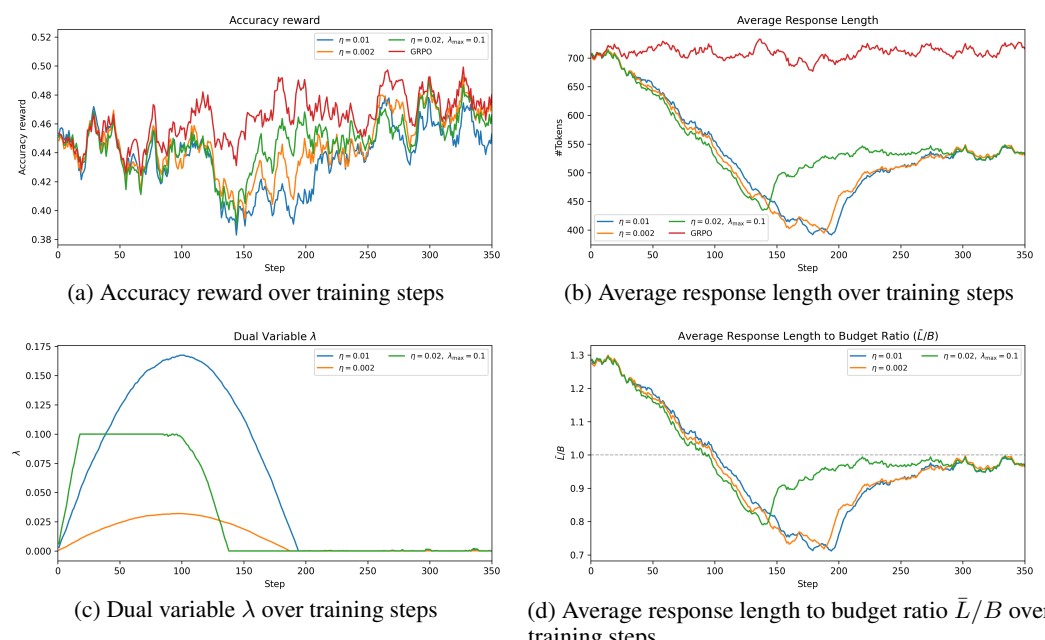

(a) Accuracy reward over training steps

(b) Average response length over training steps

(c) Dual variable $\lambda$ over training steps

(d) Average response length to budget ratio $\bar{L}/B$ over training steps

Figure 4: Ablation of the length-aware fine-tuning: (a) reward accuracy, (b) mean response length, (c) length-penalty multiplier $\lambda$, (d) ratio of average response length over threshold.

Table 4: Evaluation results across 4 math benchmarks.

| Model | AIME2024 | | MATH | | Minerva | | Olympiad | | Macro Average | |
|---|---|---|---|---|---|---|---|---|---|---|
| | Pass@1 | # Tokens | Pass@1 | # Tokens | Pass@1 | # Tokens | Pass@1 | # Tokens | Pass@1 | # Tokens |
| Qwen-Math-1.5B | | | | | | | | | | |
| + GRPO | 11.46 | 952 | 74.86 | 570 | 25.14 | 656 | 39.84 | 808 | 37.89 | 747 |
| + LACONIC (0.02) | 11.04 | 661 | 73.73 | 466 | 25.34 | 539 | 37.72 | 601 | 36.96 ↓0.93 | 567 ↓24% |
| + LACONIC (0.01) | 11.46 | 674 | 73.85 | 464 | 25.25 | 524 | 38.39 | 603 | 37.24 ↓0.65 | 566 ↓24% |
| + LACONIC (0.002) | 10.42 | 652 | 73.74 | 464 | 25.32 | 527 | 38.43 | 609 | 36.98 ↓0.91 | 563 ↓25% |

We report in figure 5 wall-clock step time, step time per token, and NVML GPU memory, averaged over training steps. LACONIC is end-to-end cheaper than vanilla RL-tuning. Our method is 19% faster and uses 22% less GPU memory. Per-token cost is nearly unchanged, with a small bookkeeping overhead for the length cost and dual update. Overall, LACONIC adds negligible

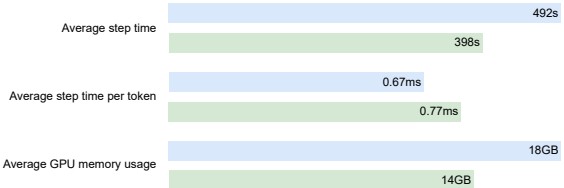

Figure 5: Average computational resource usage of LACONIC (green) and GRPO (blue).

kernel-level overhead and reduces overall runtime and memory by learning to generate fewer tokens.

## 5 RELATED WORK

**RL fine-tuning.** Reinforcement learning has become a crucial component of LLM post-training, particularly for enhancing large-scale reasoning and aligning model behavior with human preferences or task-specific objectives. One of the most widely used approaches for policy optimization in this setting is the policy gradient algorithm [Sutton & Barto, 2018]. To address the instability of early methods such as REINFORCE [Williams, 1992], Proximal Policy Optimization (PPO) [Schulman et al., 2017] introduced a clipped importance sampling ratio to constrain policy updates, which has since become a standard in RL training pipelines for LLMs. More recently, GRPO [Shao et al., 2024b] proposed group-relative advantage estimation, which eliminates the need for a learned value

function, reducing variance and improving computational efficiency. This innovation has catalyzed a wave of follow-up methods aimed at improving sample efficiency, stability, and performance in RL fine-tuning.

Several GRPO-based extensions have been introduced to address specific challenges in LLM training. SRPO [Zhang et al., 2025] addresses the problem of ineffective samples through history resampling, enhancing credit assignment in sparse-reward settings. DAPO [Yu et al., 2025] introduces dynamic sampling and a token-level gradient loss to better handle complex, multi-step reasoning tasks such as chain-of-thought (CoT) generation. Other variants include VAPO [Yue et al., 2025], which adapts advantage estimation to better capture variance across different reasoning depths; GSPO [Zheng et al., 2025], which emphasizes group-level structure in sampling; GFPO [Shrivastava et al., 2025], which focuses on sample efficiency in long-horizon settings; and GMPO [Zhao et al., 2025], which explores geometric averaging of policy gradients for improved robustness. Collectively, these methods demonstrate the growing sophistication of RL fine-tuning techniques and the community's effort to make them more scalable, stable, and effective for large-scale LLM alignment.

**Length-Aware LLMs.** Recent work has investigated various approaches for making large language models (LLMs) aware of output length, including modifications to positional encoding, prompt engineering techniques, and post-hoc truncation methods [Li et al., 2025a; Wu et al., 2025; Li et al., 2025b]. A common strategy involves incorporating length preferences into reinforcement learning (RL) fine-tuning through manually designed reward functions that penalize or incentivize certain output lengths [Aggarwal & Welleck, 2025; Cheng et al., 2025; Huang et al., 2025; Yuan et al., 2025]. These methods typically rely on fixed heuristics or penalty terms that remain constant throughout training, and thus optimize a surrogate objective that may be misaligned with the true downstream task reward. This misalignment can lead to suboptimal performance and often requires extensive hyperparameter tuning to balance length control with task-specific quality. Recent efforts have also explored dynamic decoding strategies and curriculum learning to improve length adaptation, but these approaches still depend on manually specified schedules or heuristics. Our work differs in that it aims to align length control with task reward in a more adaptive and data-driven manner, avoiding the limitations of fixed shaping objectives.

## 6 CONCLUSION

We present LACONIC, a budgeted, feedback-driven formulation for length-aware reinforcement learning (RL) fine-tuning that integrates seamlessly into standard GRPO pipelines with minimal modifications. The method introduces a zero-penalty window up to a user-specified token budget and applies a single adaptive dual variable beyond that threshold, enabling the model to align incentives for concise reasoning without compromising accuracy. Unlike prior approaches that rely on fixed heuristics or extensive hyperparameter tuning, LACONIC learns to balance task reward and response length in a principled, data-driven manner. Across a suite of reasoning and code-generation benchmarks, the method consistently reduces average output length while preserving or even improving pass@1. Controlled ablations that vary only the token budget confirm precise and stable controllability: training lengths closely track the target, and evaluated models shorten proportionally as the threshold tightens—without the need to retune the dual learning rate or other hyperparameters. Furthermore, the method generalizes across tasks with minimal tuning, demonstrating robustness and scalability. In conclusion, LACONIC provides a simple and effective mechanism to trade off quality for cost, elevating length control to a first-class, reliable component of RL-based LLM fine-tuning.

### LIMITATIONS AND FUTURE WORK.

While LACONIC is effective and lightweight, it has several limitations. It currently enforces a global average token budget, which may not capture prompt-specific or context-dependent needs. Extending to per-prompt or adaptive budgets could improve flexibility. The method also assumes a fixed, reliable reward model, which may be unrealistic in noisy or subjective tasks—robust or uncertainty-aware variants are a promising direction. Our experiments are limited to math reasoning; future work could validate generality on dialogue, summarization, or code. Finally, LACONIC handles a single constraint, but the framework naturally extends to multi-constraint settings such as latency or safety.

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

LLM USAGE

We used LLMs as general-purpose writing and debugging assistants. Specifically, LLMs were employed to help polish the writing (e.g., improving sentence clarity, grammar, and flow) and occasionally to assist with debugging minor implementation issues (e.g., identifying syntax errors or suggesting code refactoring). However, all core ideas, research questions, methodological designs, codebase implementations, experiments, and analyses were entirely conceived, developed, and conducted by the authors. No part of the intellectual contribution, experimental framework, or scientific reasoning was generated by an LLM.

APPENDIX

## A   ABLATION ANALYSIS ON COST FUNCTION

In this section, we discuss an alternative cost function

$$\widetilde{c}(q, o) = \frac{L(o) - B}{B}. \tag{7}$$

Recall the constrained optimization problem of length-aware LLMs formulated in equation (2):

$$\max_\theta \mathbb{E}_{q \sim P(Q), o \sim \pi_\theta(\cdot|q)}[r(q, o)], \text{ s.t. } \mathbb{E}_{q \sim P(Q), o \sim \pi_\theta(\cdot|q)}[L(o)] \leq B.$$

The corresponding Lagrangian is

$$\mathcal{L}(\theta, \lambda) = \mathbb{E}_{q \sim P(Q), o \sim \pi_\theta(\cdot|q)}[r(q, o)] - \lambda \big( \frac{\mathbb{E}_{q \sim P(Q), o \sim \pi_\theta(\cdot|q)}[L(o)]}{B} - 1 \big), \quad \lambda \geq 0, \tag{8}$$

and the problem equivalently translates to $\min_{\lambda \geq 0} \max_\theta \mathcal{L}(\theta, \lambda)$. The standard primal-dual approach solves the primal variable $\theta$ and dual variable $\lambda$ iteratively with partial derivatives. Specifically, this gives us the gradient-descent-style dual update rule:

$$\lambda \leftarrow \max\{\lambda + \eta \frac{\partial \mathcal{L}(\theta, \lambda)}{\partial \lambda}, 0\} = \max\{\lambda + \eta \big( \frac{\mathbb{E}_{q \sim P(Q), o \sim \pi_\theta(\cdot|q)}[L(o)]}{B} - 1 \big), 0\}. \tag{9}$$

Empirically we estimate the expectation $\mathbb{E}_{q \sim P(Q), o \sim \pi_\theta(\cdot|q)}[L(o)]$ with the minibatch mean $\bar{L}$, and we derive exactly equation (6):

$$\lambda \leftarrow \max\{\lambda + \eta(\frac{\bar{L}}{B} - 1), 0\}.$$

The immediate primal update rule is given by

$$\theta \leftarrow \arg \max_\theta \mathbb{E}_{q \sim P(Q), o \sim \pi_\theta(\cdot|q)}[r(q, o)] - \lambda \big( \frac{\mathbb{E}_{q \sim P(Q), o \sim \pi_\theta(\cdot|q)}[L(o)]}{B} - 1 \big). \tag{10}$$

By linearity of expectations and removing constants inside $\arg \max$, we have

$$\theta \leftarrow \arg \max_\theta \mathbb{E}_{q \sim P(Q), o \sim \pi_\theta(\cdot|q)}[r(q, o) - \lambda \frac{L(o) - B}{B}]. \tag{11}$$

By definition of $\widetilde{c}(q, o)$, we have

$$\theta \leftarrow \arg \max_\theta \mathbb{E}_{q \sim P(Q), o \sim \pi_\theta(\cdot|q)}[r(q, o) - \lambda \cdot \widetilde{c}(q, o)]. \tag{12}$$

Therefore, the linear cost function $\widetilde{c}(q, o)$ serves as a natural cost function for our constrained RL formulation.

However, using the purely linear cost function $\widetilde{c}(q, o) = \frac{L(o) - B}{B}$ inside the primal (actor model) update in equation (12) is problematic in practice. Whenever $\lambda > 0$, the actor model is incentivized to shorten responses on all samples, and collapses the policy to extremely short outputs. This leads to highly unstable training dynamics, which is undesirable in practice. We conduct ablation experiments

Table 5: Ablation results on linear cost functions.

| Model | AIME2024 | | MATH | | Minerva | | Olympiad | | Macro Average | |
| --- | --- | --- | --- | --- | --- | --- | --- | --- | --- | --- |
| | Pass@1 | # Tokens | Pass@1 | # Tokens | Pass@1 | # Tokens | Pass@1 | # Tokens | Pass@1 | # Tokens |
| Qwen-Math-1.5B | | | | | | | | | | |
| **+ LACONIC** | 11.46 | 674 | 73.85 | 464 | 25.25 | 524 | 38.39 | 603 | 37.24 | 566 |
| + linear costs | 11.35 | 665 | 73.42 | 479 | 25.53 | 531 | 38.42 | 627 | 37.18 | 576 |

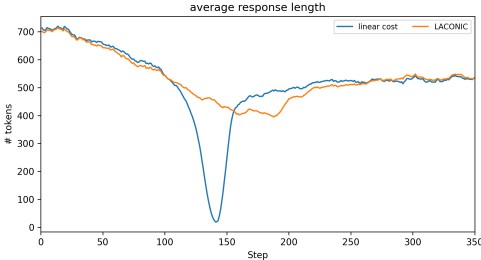

(a) Accuracy reward over training steps    (b) Average response length over training steps

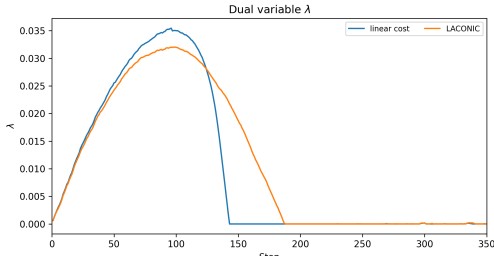

(c) Dual variable $\lambda$ over training steps

Figure 6: Ablation of the cost functions: (a) reward accuracy, (b) mean response length, (c) length-penalty multiplier $\lambda$,

by replacing the cost function with $\widetilde{c}$ while keep all other settings the same. We present the evaluation results of resulting checkpoints in table 5.

We also plot the training dynamics in figure 6. The experiment shows that with linear cost function $\frac{L(o)-B}{B}$, the average accuracy plummets from 40% to 8%, and the mean response lengths fall below 10 tokens, indicating that the model is no longer producing meaningful responses during training. Although as shown in table 5, the primal-dual framework with linear cost function $\widetilde{c}$ recovers the model's performance after the model is stabilized, the unstable update steps can introduce great risk and instability to LLM fine-tuning.

# B TRAINING DETAILS

## B.1 TRAINING SETUP

For the main results, we train DeepScaleR-1.5B with a 2000-token budget for 150 steps, then further train the saved checkpoint with a 1000-token budget for another 100 steps. We train Qwen-Math-1.5B with a 550-token budget for 350 steps. We set batch size to 128, lambda initialized to 0, lambda ceiling to 0.03, step size $\eta$ to 0.002, and rollout number to 7.

## B.2 TRAINING DYNAMICS

We present the training dynamics of the clip ratio in appendix B.2. The logged data of clip ratio show that 35-40% responses generated by GRPO are clipped, while LACONIC is able to control the clip ratio under 10%. This partially explains the "counterintuitive" results where LACONIC yields better

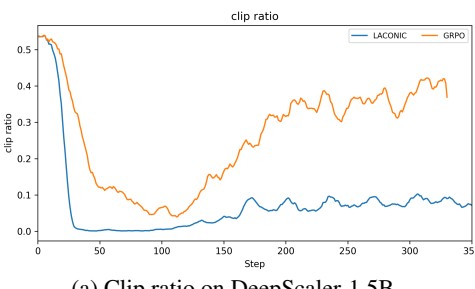 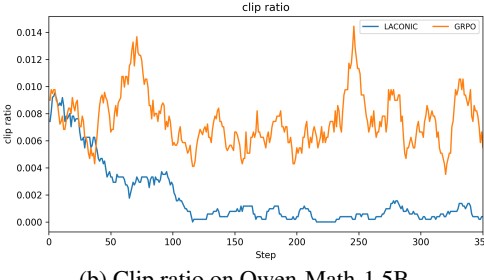

(a) Clip ratio on DeepScaler-1.5B.    (b) Clip ratio on Qwen-Math-1.5B.

Figure 7: Training dynamics of clip ratio. The clip ratio is the ratio of clipped responses in the current minibatch, and a response is clipped when its length reaches the maximum response length, in our experiment, 4096 tokens for DeepScaler-1.5B and 2048 for Qwen-Math-1.5B.

pass@1 on many benchmarks than GRPO, despite the token budget constraints, as high clip-ratio of GRPO causes insufficient training on corresponding prompts.

## C  CASE STUDY

Prompt: Every morning Aya goes for a 9-kilometer-long walk and stops at a coffee shop afterwards. When she walks at a constant speed of $s$ kilometers per hour, the walk takes her 4 hours, including $t$ minutes spent in the coffee shop. When she walks $s + 2$ kilometers per hour, the walk takes her 2 hours and 24 minutes, including $t$ minutes spent in the coffee shop. Suppose Aya walks at $s + \frac{1}{2}$ kilometers per hour. Find the number of minutes the walk takes her, including the $t$ minutes spent in the coffee shop. You FIRST think about the reasoning process as an internal monologue and then provide the final answer. The reasoning process MUST BE enclosed within <think> </think> tags. The final answer MUST BE put in \boxed{}.

Ground truth answer: $204$.

> **LACONIC Output (#Tokens = 855)**
>
> Alright, let's tackle this problem step by step.
> First, let's understand what's given. Aya walks a total distance of 9 kilometers. The time she takes depends on her walking speed. She stops at a coffee shop, and the time spent there is $t$ minutes.
> We are given two scenarios:
> 1. When her speed is $s$ km/h, the total time (walking plus coffee) is 4 hours. 2. When her speed increases to $s + 2$ km/h, the total time becomes 2 hours and 24 minutes, which is $2 + \frac{24}{60} = 2.4$ hours.
> We need to find the total time when her speed is $s + 0.5$ km/h.
> Let's denote the time spent in the coffee shop as $t$ minutes. Since the coffee shop time is the same in both scenarios, we can set up equations based on the total time.
> In the first scenario: Total time = walking time + $t$ minutes. Walking time = distance / speed $= 9/s$ hours. Therefore, $9/s + t = 4$ hours.
> In the second scenario: Walking time = $9/(s + 2)$ hours. Total time = $9/(s + 2) + t = 2.4$ hours.
> Now, we have two equations: 1. $9/s + t = 4$ 2. $9/(s + 2) + t = 2.4$
> We can subtract equation 2 from equation 1 to eliminate $t$:
> $(9/s + t) - (9/(s + 2) + t) = 4 - 2.4$
> Simplify: $9/s - 9/(s + 2) = 1.6$
> Factor out 9: $9[1/s - 1/(s + 2)] = 1.6$
> Compute the difference inside the brackets:
> $1/s - 1/(s + 2) = (s + 2 - s)/(s(s + 2)) = 2/(s(s + 2))$ So, $9 * (2)/(s(s + 2)) = 1.6$
> Simplify: $18/(s(s + 2)) = 1.6$

Multiply both sides by $s(s + 2)$: $18 = 1.6\, s(s + 2)$
Divide both sides by 1.6: $s(s + 2) = 18/1.6$
Calculate $18/1.6$: $18/1.6 = 11.25$
So, $s^2 + 2s - 11.25 = 0$
Now, solve the quadratic equation: $s^2 + 2s - 11.25 = 0$
Using quadratic formula: $s = [-2 \pm \sqrt{4 + 45}]/2$
Since speed can't be negative: $s = [-2 + \sqrt{49}]/2 \; \sqrt{49} = 7$
Thus, $s = (-2 + 7)/2 = 5/2 = 2.5$ km/h
Now, we can find $t$ from equation 1: $9/s + t = 4 \; 9/2.5 + t = 4$
Compute $9/2.5 = 3.6$ So, $3.6 + t = 4$
Thus, $t = 0.4$ hours $= 0.4 * 60 = 24$ minutes.
Now, when her speed is $s + 0.5 = 2.5 + 0.5 = 3$ km/h.
We need to find the total time including coffee shop.
First, compute walking time: $9/3 = 3$ hours.
Total time = walking time + coffee shop time $= 3 + 0.4 = 3.4$ hours.
Convert to minutes: $3.4 * 60 = 204$ minutes.
So, the total time is 204 minutes.
</think>
The total time Aya takes when walking at $s + \frac{1}{2}$ km/h is $\boxed{204}$ minutes.

---

**DSR-4K (#Tokens = 1763)**

Okay, so I have this problem about Aya's walking and coffee shop time. Let me try to figure it out step by step.

First, the problem says that Aya walks 9 kilometers every morning and stops at a coffee shop. When she walks at a speed of $s$ km/h, it takes her 4 hours, including $t$ minutes spent in the coffee shop. Then, when she increases her speed to $s + 2$ km/h, the walk takes her 2 hours and 24 minutes, but this time includes the same $t$ minutes in the coffee shop.

I need to find the time it takes for Aya to walk at a speed of $s + \frac{1}{2}$ km/h, including the $t$ minutes in the coffee shop.

Let me break down the information given:

1. When walking at speed $s$ km/h, the total time taken is 4 hours, including $t$ minutes in the coffee shop. So, the walking time is 4 hours minus $t$ minutes. But since speed is in km/h, maybe it's better to convert everything into hours.

2. When walking at speed $s + 2$ km/h, the total time is 2 hours and 24 minutes, which is $2 + \frac{24}{60} = 2.4$ hours. Again, the walking time is total time minus $t$ minutes.

So, let me denote:

- Walking speed: $s$ km/h
- Walking speed in the second case: $s + 2$ km/h
- Total time in first case: 4 hours
- Total time in second case: 2.4 hours
- Both cases include $t$ minutes in the coffee shop.

We need to find the total time when walking speed is $s + \frac{1}{2}$ km/h.
First, let's convert the walking times into hours to make calculations easier.
For the first scenario:
Total time = 4 hours
Time spent walking $= 4 - \frac{t}{60}$ hours (since $t$ is in minutes)
Distance walked is 9 km, so using the formula:
Distance = speed × time
So, $9 = s \times \left(4 - \frac{t}{60}\right)$
Similarly, for the second scenario:
Total time = 2.4 hours
Time spent walking $= 2.4 - \frac{t}{60}$ hours
Distance walked is 9 km again:

$9 = (s + 2) \times \left(2.4 - \frac{t}{60}\right)$

Now, we have two equations:

1) $9 = s \times \left(4 - \frac{t}{60}\right)$

2) $9 = (s + 2) \times \left(2.4 - \frac{t}{60}\right)$

Since both equal 9, we can set them equal to each other:

$s \times \left(4 - \frac{t}{60}\right) = (s + 2) \times \left(2.4 - \frac{t}{60}\right)$

Let me expand both sides:

Left side: $4s - s\left(\frac{t}{60}\right)$

Right side: $(s + 2)\left(2.4 - \frac{t}{60}\right) = 2.4s - s\left(\frac{t}{60}\right) + 4.8 - 2\left(\frac{t}{60}\right)$

Simplify both sides:

Left: $4s - \dfrac{st}{60}$

Right: $2.4s - \dfrac{st}{60} + 4.8 - \dfrac{2t}{60}$

Now, let's write the equation:

$4s - \dfrac{st}{60} = 2.4s - \dfrac{st}{60} + 4.8 - \dfrac{2t}{60}$

Notice that the $-\dfrac{st}{60}$ terms on both sides cancel each other out. So we can subtract $2.4s$ from both sides:

$4s - 2.4s = -\dfrac{st}{60} + \dfrac{st}{60} + 4.8 - \dfrac{2t}{60}$

Simplify:

$1.6s = 4.8 - \dfrac{2t}{60}$

Simplify $1.6s = 4.8 - \dfrac{t}{30}$

Let me write that as:

$1.6s + \dfrac{t}{30} = 4.8$

**Equation (A):** $1.6s + \dfrac{t}{30} = 4.8$

Now, let's go back to the first equation:

$9 = s \times \left(4 - \frac{t}{60}\right)$

Let me write this as:

$9 = 4s - \dfrac{st}{60}$

Multiply both sides by 60 to eliminate denominators:

$9 \times 60 = 4s \times 60 - st$

$540 = 240s - st$

Let's write this as:

$st = 240s - 540$

Divide both sides by $s$ (assuming $s \neq 0$):

$t = 240 - \dfrac{540}{s}$

**Equation (B):** $t = 240 - \dfrac{540}{s}$

Now, substitute Equation (B) into Equation (A):

$1.6s + \dfrac{240 - \frac{540}{s}}{30} = 4.8$

Simplify:

$1.6s + \dfrac{240}{30} - \dfrac{540}{30s} = 4.8$

Calculate:

$1.6s + 8 - \dfrac{18}{s} = 4.8$

Now, subtract 4.8 from both sides:

$1.6s + 8 - \dfrac{18}{s} - 4.8 = 0$

Simplify:

$$1.6s + 3.2 - \frac{18}{s} = 0$$

Multiply both sides by $s$ to eliminate denominator:

$$1.6s^2 + 3.2s - 18 = 0$$

Now, we have a quadratic equation:

$$1.6s^2 + 3.2s - 18 = 0$$

Let me multiply both sides by 10 to eliminate decimals:

$$16s^2 + 32s - 180 = 0$$

Divide all terms by 4 to simplify:

$$4s^2 + 8s - 45 = 0$$

Now, solve for $s$ using quadratic formula:

$$s = \frac{-b \pm \sqrt{b^2 - 4ac}}{2a}$$

Where $a = 4$, $b = 8$, $c = -45$

Calculate discriminant:

$$D = 8^2 - 4 \times 4 \times (-45) = 64 + 720 = 784$$

Square root of $D$: $\sqrt{784} = 28$

So,

$$s = \frac{-8 \pm 28}{2 \times 4} = \frac{-8 \pm 28}{8}$$

We have two solutions:

1) $s = \dfrac{-8 + 28}{8} = \dfrac{20}{8} = 2.5$ km/h

2) $s = \dfrac{-8 - 28}{8} = \dfrac{-36}{8} = -4.5$ km/h

Since speed cannot be negative, $s = 2.5$ km/h

Now, find $t$ using Equation (B):

$$t = 240 - \frac{540}{s} = 240 - \frac{540}{2.5}$$

Calculate $\dfrac{540}{2.5}$:

$$540/2.5 = 216$$

So,

$$t = 240 - 216 = 24 \text{ minutes}$$

So, $t = 24$ minutes.

Now, when Aya walks at $s + \frac{1}{2}$ km/h, which is $2.5 + 0.5 = 3$ km/h.

We need to find the total time including $t$ minutes in the coffee shop.

First, calculate the walking time at 3 km/h.

Distance is 9 km, speed is 3 km/h, so time is $\dfrac{9}{3} = 3$ hours.

But total time is walking time plus $t$ minutes.

Walking time is 3 hours, which is 180 minutes.

Plus $t = 24$ minutes.

So total time is $180 + 24 = 204$ minutes.

Therefore, the walk takes her 204 minutes, including the $t$ minutes spent in the coffee shop.

</think>

The number of minutes the walk takes her, including the $t$ minutes spent in the coffee shop, is

$\boxed{204}$.

