# OpenReview forum: "LACONIC: Length-Aware Constrained Reinforcement Learning for LLM"
_ICLR.cc/2026/Conference — Submitted to ICLR 2026_

### Official Review · Reviewer_kveD · 2025-10-29

**Soundness:** 2
**Presentation:** 2
**Contribution:** 2
**Rating:** 4
**Confidence:** 4

**Summary:**

The paper introduces LACONIC, a reinforcement learning (RL) method designed to control the output length of large language models (LLMs) during fine-tuning. The authors identify that RL-tuned LLMs often generate verbose reasoning traces that inflate inference cost and latency. To address this, LACONIC reformulates RL fine-tuning as a constrained optimization problem that maximizes task reward while enforcing an average token budget constraint. Empirical results across multiple reasoning benchmarks demonstrate that LACONIC achieves similar or better accuracy compared to standard RL methods like GRPO. The paper further validates LACONIC's generalization to out-of-domain tasks and shows ablations confirming robustness to hyperparameters and computational efficiency improvements.

**Strengths:**

- The proposed solution is interesting. The method elegantly reinterprets length control as a constrained optimization problem rather than heuristic reward shaping.
- The paper is easy to follow.

**Weaknesses:**

- The major concern is that the experiments are only conducted on 1.5B-scale models. Also, the results of Qwen2.5-Math-1.5B-Instruct are not very strong compared with GRPO.
- While the primal-dual approach is conceptually sound, the paper lacks formal convergence analysis.
- Baselines are mainly GRPO and L1-based methods. It would strengthen the paper to include comparisons with more recent efficiency-oriented RL approaches (e.g., GFPO [1]).

[1] Sample More to Think Less: Group Filtered Policy Optimization for Concise Reasoning.

**Questions:**

Besides the weaknesses above, further questions are as follows:

- Why does LACONIC (3000) have higher performance than LACONIC (2000) in Table 3?

---

> ### Author Response · Authors · 2025-11-23
> **Authors' Responses (Part 1)**
>
> We thank the reviewer for the effort to review our work and for providing constructive feedback and insightful comments for our work. We respond to the reviewer's concerns below.
>
> > Comment 1: The major concern is that the experiments are only conducted on 1.5B-scale models.
>
> We thank the reviewer for raising this point. We add experiments on the Qwen3-8B model and present the results in the following table:
>
> **Table 1: Experiment results on Qwen3-8B.**
>
> | Model                 | AIME2024 |          | MATH  |          | Minerva |          | Olympiad |          | Macro Avg |          |
> |-----------------------|----------|----------|-------|----------|---------|----------|----------|----------|-----------|----------|
> |                       | Pass@1   | #Tokens | Pass@1 | #Tokens | Pass@1 | #Tokens | Pass@1  | #Tokens | Pass@1    | #Tokens |
> | Qwen3-8B     |                 |                   |             |                |                 |                  |                  |                   |                   |                    |
> | + GRPO       | 37.50           | 2918              | 87.95       | 1532           | 40.14           | 1549             | 54.40            | 2167              | 55.00             | 2042               |
> | **+ `LACONIC`** | 30.42        | 1643              | 87.01       | 792            | 38.46           | 783              | 53.05            | 1166              | 52.23             | 1096               |
>
> In the LACONIC training experiment, we set the token budget $B=1100$. The results show that LACONIC achieves comparable pass@1 compared to the GRPO baseline with a 2.73\% drop, while significantly reduce the response lengths by 46\%. This shows that LACONIC can effectively shorten response lengths while preserving reasoning abilities on large reasoning models.
>
> ---
>
> > Comment 2: The results of Qwen2.5-Math-1.5B-Instruct are not very strong compared with GRPO.
>
> We would like to clarify that Qwen2.5-Math-1.5B-Instruct is an instruction-tuned model with standard chain-of-thought training, in contrast to R1-style long-CoT models such as DeepScaleR-1.5B-Preview. Although the relative shortening in response length is more moderate in this setting (24\% vs. 53\%, cf. Table 1 in the paper and the updated evaluation results below), Qwen2.5-Math-1.5B-Instruct already produces relatively short solutions (747 tokens on average). In this regime, further compressing the responses without degrading accuracy is substantially more challenging. Our experiments on Qwen2.5-Math-1.5B-Instruct therefore illustrate a complementary aspect of LACONIC. Even when starting from a "short" CoT model, LACONIC can still yield additional reduction in response length while essentially preserving the model's reasoning performance.
>
> ---
>
> > Comment 3: While the primal-dual approach is conceptually sound, the paper lacks formal convergence analysis.
>
> We agree that a formal convergence analysis would further strengthen the paper. At the same time, deriving such guarantees in our setting is technically nontrivial. Our primal updates use a ReLU-style cost transformation to stabilize the policy updates, which falls outside the standard primal-dual convergence analysis. Developing a new theoretical framework to handle this ReLU-based update rule is an interesting direction, but it is orthogonal to the main scope of this work, which is to demonstrate our length-aware primal-dual approach can effectively control response lengths while preserving task performance through empirical studies. We believe the extensive experiments across models and benchmarks provide strong evidence for the practical effectiveness and stability of our method. We will add a brief discussion of the theoretical limitation and future work in the final version.

---

> ### Author Response · Authors · 2025-11-23
> **Authors' Responses (Part 2)**
>
> > Comment 4: Baselines are mainly GRPO and L1-based methods. It would strengthen the paper to include comparisons with more recent efficiency-oriented RL approaches (e.g., GFPO [1]).
>
> We thank the reviewer for the constructive feedback. We agree that incorporating more recent efficiency-oriented RL approaches as baselines can further strengthen the paper. We have added ThinkPrune [2] as an additional efficiency-oriented baseline, as suggested by Reviewer 9Kz2. The authors of [2] fine-tune DeepScaleR-1.5B-Preview with two strategies (ThinkPrune-2k and ThinkPrune-Iter2k) and release the resulting checkpoints on HuggingFace. We evaluate these public checkpoints on AIME2024, MATH, Minerva, and Olympiad Bench under a unified evaluation protocol: maximum context length 32768 and average over 16 samples with top\_p = 0.95 and temperature = 0.6.
>
> **Table 2: Updated main experiment results.**
>
> | Model                 | AIME2024 |          | MATH  |          | Minerva |          | Olympiad |          | Macro Avg |          |
> |-----------------------|----------|----------|-------|----------|---------|----------|----------|----------|-----------|----------|
> |                       | Pass@1   | #Tokens | Pass@1 | #Tokens | Pass@1 | #Tokens | Pass@1  | #Tokens | Pass@1    | #Tokens |
> | DeepScaleR-1.5B   | 38.75           | 8140              | 85.92       | 3019           | 27.62           | 4134             | 51.94            | 5410              | 51.06             | 5176               |
> | + GRPO                | 29.58    | 6122     | 85.28 | 1767     | 29.64  | 2630     | 49.11   | 3418     | 48.40     | 3484     |
> | + L1-Exact            | 22.02    | 4138     | 82.57 | 3734     | 28.74  | 3755     | 44.91   | 3987     | 44.56     | 3904     |
> | + L1-Max              | 25.12    | 2979     | 83.45 | 1912     | 28.35  | 1701     | 44.76   | 2334     | 45.42     | 2231     |
> | + ThinkPrune2k        | 30.42    | 4275     | 83.52 | 1854     | 29.87  | 1961     | 46.57   | 2805     | 47.59     | 2724     |
> | + ThinkPrune-Iter2k   | 35.00    | 4598     | 84.22 | 1802     | 30.85  | 1941     | 48.77   | 2935     | 49.71     | 2819     |
> | **+ `LACONIC`**       | 28.12    | 2665     | 83.75 | 1049     | 29.24  | 1189     | 47.07   | 1669     | 47.05     | 1643     |
>
> Given that [1] has not released code or checkpoints, and lacks sufficient details for reliable reproduction, we respectfully omit it as a baseline for now and will briefly discuss GFPO in related work and limitations.
>
> ---
>
> We answer the reviewer's question below.
>
> > Question 1: Why does LACONIC (2000) have higher performance than LACONIC (3000) in Table 3?
>
> We thank the review for raising this good question. In these experiments, we set the maximum response length to be 4096. The 3000 token budget is hardly restrictive, where standard GRPO can naturally satisfy the constraint, and with LACONIC, $\lambda$ drops to 0 in early stages (c.f. Figure 3 (b),(c) in the paper). Due to the lack of model updates toward shorter responses, many responses reach the 4096 length cap and are clipped, leading to insufficient training on these prompts.
>
> ---
>
> Reference:
>
> [1] Shrivastava et al. Sample More to Think Less: Group Filtered Policy Optimization for Concise Reasoning, 2025. URL: https://arxiv.org/abs/2508.09726.
>
> [2] Hou et al. ThinkPrune: Pruning Long Chain-of-Thought of LLMs via Reinforcement Learning, 2025. URL: https://arxiv.org/abs/2504.01296.

---

> ### Author Response · Authors · 2025-12-03
> **Authors' Responses (Part 3)**
>
> We additionally evaluate LACONIC on DeepSeek-R1-Distill-Qwen-1.5B. We fine-tune the base model with a token budget $B=1500$ for 500 RL steps. At evaluation time, we compare our checkpoint against the base model, ThinkPrune-Iter2k [1], and Efficient-Reasoning [2] on AIME2024, MATH, Minerva, and Olympiad. The maximum response length is set to 32,768 during evaluation. The results are reported in Table 3.
>
> Table 3: Results on DeepSeek-R1-1.5B.
>
> | Model                 | AIME2024 |          | MATH  |          | Minerva |          | Olympiad |          | Macro Avg |          |
> |-----------------------|----------|----------|-------|----------|---------|----------|----------|----------|-----------|----------|
> |                       | Pass@1   | #Tokens | Pass@1 | #Tokens | Pass@1 | #Tokens | Pass@1  | #Tokens | Pass@1    | #Tokens |
> | DeepSeek-R1-1.5B      | 30.00           | 15218             | 82.90       | 5340          | 29.13          | 6783             | 44.8            | 12417             | 46.71        | 9940           |
> | + Efficient-Reasoning | 22.08           | 8801              | 81.63       | 1909          | 27.57          | 2018             | 42.96           | 4799              | 43.56        | 4382           |
> | + ThinkPrune-Iter2k   | 23.33           | 5440              | 81.35       | 1762          | 27.35          | 1837             | 43.39           | 3021              | 43.85        | 3015           |
> | **+ `LACONIC`**         | 24.17       | 5140          | 81.88   | 1636      | 28.36      | 1638         | 44.11       | 3241          | 44.63    | 2914       |
>
> These results show that LACONIC consistently achieves higher pass@1 with fewer tokens than existing length-control methods [1,2] on DeepSeek-R1-Distill-Qwen-1.5B. Relative to the original base model, LACONIC incurs only a modest macro-average performance drop of 2.08 percentage points, while using approximately 71\% fewer tokens on average.
>
> ---
>
> References
>
> [1] Hou et al. ThinkPrune: Pruning Long Chain-of-Thought of LLMs via Reinforcement Learning, 2025. URL: https://arxiv.org/abs/2504.01296.
>
> [2] Daman Arora and Andrea Zanette. Training Language Models to Reason Efficiently, 2025. URL: https://arxiv.org/abs/2502.04463

---

### Official Review · Reviewer_9Kz2 · 2025-11-01

**Soundness:** 2
**Presentation:** 2
**Contribution:** 2
**Rating:** 4
**Confidence:** 3

**Summary:**

This paper introduces a new method, LACONIC, which aims to improve RL-based reasoning algorithms like GRPO by enforcing target token budgets during training. The method is a primal-dual method that operates by first optimizing the policy followed by an update of the  dual variable \lambda. Experiments across math reasoning benchmarks like AIME'24, MATH500 and general reasoning like GPQA, MMLU show that the method improves 1.5B base models by matching the performance of GRPO-variants while substantially cutting down response lengths, thereby reducing inference time costs.

**Strengths:**

- The paper is well written and presented.
- The method is well motivation: the problem of inference time costs increasingly largely due to growing response lengths is important, and the paper tackles it with an appropriate solution.
- The proposed LACONIC method which introduces a primal-dual optimization strategy is technically novel.

**Weaknesses:**

- All the conducted experiments are on small-scale models (1.5B models). Previous works like Sober reasoning (https://arxiv.org/abs/2504.07086) have shown that RL on small-scale models might not be reliable. Further experiments on larger and more diverse models (7B or larger) are required to ensure that the results are conclusive and transfer to real-world scales.
- The numbers in table 1 are categorically lower than the baseline numbers reported in the Sober Reasoning paper. Since this previous work suggests that strong tuning of baselines is important for making correct claims about the gains of RL, it is imperative to follow correct baseline reporting. It would be good and important to re-evaluate all the model checkpoints in tab 1 under the fair evaluation strategy proposed by that paper to convincingly show that the results hold strongly also under fair baselining.
- Some important baselines that also aim to shorten the response lengths are missing: https://arxiv.org/pdf/2502.04463, https://arxiv.org/pdf/2504.01296. This paper should either add results comparing against these baseline methods or add a discussion regarding why these baseline comparisons are not valid / required.
- The conclusion section claims that the LACONIC method can be flexibly used across RL algorithms like PPO / GRPO etc, however the paper only shows results for GRPO. Does the method indeed improve other RL algorithms like GSPO, PPO etc? If there are no additional experiments to back this up, the claim in the conclusion must be toned down.

**Questions:**

All my questions are in the weaknesses section

---

> ### Author Response · Authors · 2025-11-23
> **Authors Responses (Part 1)**
>
> We thank the reviewer for the effort to review our work and for providing constructive feedback and insightful comments for our work. We respond to the reviewer's concerns below.
>
> > Comment 1: All the conducted experiments are on small-scale models (1.5B models). Further experiments on larger and more diverse models (7B or larger) are required to ensure that the results are conclusive and transfer to real-world scales.
>
> We thank the reviewer for raising this point. We add experiments on the Qwen3-8B model and present the results in the following table:
> **Table 1: Experiment results on Qwen3-8B.**
>
> | Model                 | AIME2024 |          | MATH  |          | Minerva |          | Olympiad |          | Macro Avg |          |
> |-----------------------|----------|----------|-------|----------|---------|----------|----------|----------|-----------|----------|
> |                       | Pass@1   | #Tokens | Pass@1 | #Tokens | Pass@1 | #Tokens | Pass@1  | #Tokens | Pass@1    | #Tokens |
> | Qwen3-8B     |                 |                   |             |                |                 |                  |                  |                   |                   |                    |
> | + GRPO       | 37.50           | 2918              | 87.95       | 1532           | 40.14           | 1549             | 54.40            | 2167              | 55.00             | 2042               |
> | **+ `LACONIC`** | 30.42        | 1643              | 87.01       | 792            | 38.46           | 783              | 53.05            | 1166              | 52.23             | 1096               |
>
> In the LACONIC training experiment, we set the token budget $B=1100$. The results show that LACONIC achieves comparable pass@1 compared to the GRPO baseline with a 2.73\% drop, while significantly reduce the response lengths by 46\%. This shows that LACONIC can effectively shorten response lengths while preserving reasoning abilities on large reasoning models.
>
> ---
>
> > Comment 2: The numbers in table 1 are categorically lower than the baseline numbers reported in the Sober Reasoning paper. It would be good and important to re-evaluate all the model checkpoints in tab 1 under the fair evaluation strategy proposed by that paper to convincingly show that the results hold strongly also under fair baselining.
>
> There are two reasons why the evaluation results of the GRPO baselines on DeepScaleR-1.5B-Preview are lower than those reported in [2]. (1) We originally set the maximum response length to 8K during evaluation to keep in line with the previous work [1] that we use as baselines, while the maximum response length is set to 32K in [2]. (2) In the GRPO baseline experiment, we train the base model
> on GRPO with maximum response length set to 4K, same as in LACONIC experiments. This is to
> satisfy the computing resource limit, and also to keep in line with previous works [1,2]. As the base model DeepScaleR-1.5B-Preview supports up to 32,768 tokens, training with the 4K length cap will cause the performance to drop.
>
> We would like to remark that the settings are identical for all experiments to ensure fair comparisons. We agree that it is more appropriate to set the maximum response length to 32K during evaluation to fully assess the model's abilities. Hence, we present the updated evaluation results as follows:
>
> **Table 2: Updated main experiment results.**
>
> | Model                 | AIME2024 |          | MATH  |          | Minerva |          | Olympiad |          | Macro Avg |          |
> |-----------------------|----------|----------|-------|----------|---------|----------|----------|----------|-----------|----------|
> |                       | Pass@1   | #Tokens | Pass@1 | #Tokens | Pass@1 | #Tokens | Pass@1  | #Tokens | Pass@1    | #Tokens |
> | DeepScaleR-1.5B   | 38.75           | 8140              | 85.92       | 3019           | 27.62           | 4134             | 51.94            | 5410              | 51.06             | 5176               |
> | + GRPO                | 29.58    | 6122     | 85.28 | 1767     | 29.64  | 2630     | 49.11   | 3418     | 48.40     | 3484     |
> | + L1-Exact            | 22.02    | 4138     | 82.57 | 3734     | 28.74  | 3755     | 44.91   | 3987     | 44.56     | 3904     |
> | + L1-Max              | 25.12    | 2979     | 83.45 | 1912     | 28.35  | 1701     | 44.76   | 2334     | 45.42     | 2231     |
> | + ThinkPrune2k        | 30.42    | 4275     | 83.52 | 1854     | 29.87  | 1961     | 46.57   | 2805     | 47.59     | 2724     |
> | + ThinkPrune-Iter2k   | 35.00    | 4598     | 84.22 | 1802     | 30.85  | 1941     | 48.77   | 2935     | 49.71     | 2819     |
> | **+ `LACONIC`**       | 28.12    | 2665     | 83.75 | 1049     | 29.24  | 1189     | 47.07   | 1669     | 47.05     | 1643     |
>
> Direct evaluation of the base model DeepScaleR-1.5B-Preview yields virtually the same pass@1 as reported in [2]. We would like to remark that our evaluation strategies are fair and sound for all models.

---

> ### Author Response · Authors · 2025-11-23
> **Authors' Responses (Part 2)**
>
> > Comment 3: Some important baselines that also aim to shorten the response lengths are missing: https://arxiv.org/pdf/2502.04463, https://arxiv.org/pdf/2504.01296. This paper should either add results comparing against these baseline methods.
>
> We thank the reviewer for these suggestions. We agree that the listed works [3,4] are relevant for length-controlled reasoning, and including them as baselines would further strengthen the paper.
>
> For [3], the authors have released DeepScaleR-1.5B-Preview checkpoints on HuggingFace, which allows a direct comparison. We have  evaluated the ThinkPrune-2k and ThinkPrune-Iter2k variants and report the results in the updated table above. Our method LACONIC attains comparable pass@1 to ThinkPrune-2k and ThinkPrune-Iter2k (within 0.54\% and 2.66\% respectively), while using 40\% and 42\% fewer tokens. Given that [4] has not released checkpoints on our base models and lacks sufficient details for reliable reproduction, we respectfully omit it as a baseline for now and will briefly discuss [4] in related work and limitations.
>
> ---
>
> > Comment 4: The conclusion section claims that the LACONIC method can be flexibly used across RL algorithms like PPO / GRPO etc, however the paper only shows results for GRPO.
>
> We thank the reviewer for raising this point. We agree that while LACONIC is conceptually compatible with other policy-gradient methods such as PPO/GSPO, our current experiments only provide empirical evidence for its effectiveness when deployed with GRPO. We will revise the conclusion to avoid over-claiming. Specifically, we will restrict the statement to our GRPO-based setup and clarify that extending and validating LACONIC under other RL algorithms (e.g., PPO/GSPO) is an interesting direction for future work.
>
> ---
>
> Reference:
>
> [1] Pranjal Aggarwal and Sean Welleck. L1: Controlling how long a reasoning model thinks with
> reinforcement learning, 2025. URL: https://arxiv.org/abs/2503.04697.
>
> [2] Hochlehnert et al. A Sober Look at Progress in Language Model Reasoning: Pitfalls and Paths to Reproducibility, 2025. URL: https://arxiv.org/abs/2504.07086
>
> [3] Hou et al. ThinkPrune: Pruning Long Chain-of-Thought of LLMs via Reinforcement Learning, 2025. URL: https://arxiv.org/abs/2504.01296.
>
> [4] Daman Arora and Andrea Zanette. raining Language Models to Reason Efficiently, 2025. URL: https://arxiv.org/abs/2502.04463

---

> ### Author Response · Authors · 2025-12-03
> **Authors' Response (Part 3)**
>
> We additionally evaluate LACONIC on DeepSeek-R1-Distill-Qwen-1.5B. We fine-tune the base model with a token budget $B=1500$ for 500 RL steps. At evaluation time, we compare our checkpoint against the base model, ThinkPrune-Iter2k [1], and Efficient-Reasoning [2] on AIME2024, MATH, Minerva, and Olympiad. The maximum response length is set to 32,768 during evaluation. The results are reported in Table 3.
>
> Table 3: Results on DeepSeek-R1-1.5B.
>
> | Model                 | AIME2024 |          | MATH  |          | Minerva |          | Olympiad |          | Macro Avg |          |
> |-----------------------|----------|----------|-------|----------|---------|----------|----------|----------|-----------|----------|
> |                       | Pass@1   | #Tokens | Pass@1 | #Tokens | Pass@1 | #Tokens | Pass@1  | #Tokens | Pass@1    | #Tokens |
> | DeepSeek-R1-1.5B      | 30.00           | 15218             | 82.90       | 5340          | 29.13          | 6783             | 44.8            | 12417             | 46.71        | 9940           |
> | + Efficient-Reasoning | 22.08           | 8801              | 81.63       | 1909          | 27.57          | 2018             | 42.96           | 4799              | 43.56        | 4382           |
> | + ThinkPrune-Iter2k   | 23.33           | 5440              | 81.35       | 1762          | 27.35          | 1837             | 43.39           | 3021              | 43.85        | 3015           |
> | **+ `LACONIC`**         | 24.17       | 5140          | 81.88   | 1636      | 28.36      | 1638         | 44.11       | 3241          | 44.63    | 2914       |
>
> These results show that LACONIC consistently achieves higher pass@1 with fewer tokens than existing length-control methods [1,2] on DeepSeek-R1-Distill-Qwen-1.5B. Relative to the original base model, LACONIC incurs only a modest macro-average performance drop of 2.08 percentage points, while using approximately 71\% fewer tokens on average.
>
> ---
>
> References
>
> [1] Hou et al. ThinkPrune: Pruning Long Chain-of-Thought of LLMs via Reinforcement Learning, 2025. URL: https://arxiv.org/abs/2504.01296.
>
> [2] Daman Arora and Andrea Zanette. Training Language Models to Reason Efficiently, 2025. URL: https://arxiv.org/abs/2502.04463

---

### Official Review · Reviewer_9Fv6 · 2025-11-03

**Soundness:** 2
**Presentation:** 3
**Contribution:** 3
**Rating:** 4
**Confidence:** 4

**Summary:**

This paper proposes a method, Laconic, to limit the output length of LLMs during training on reasoning problems. Laconic formulates the length restriction as a constraint imposed upon the usual reward maximization objective. The model first performs an update in the direction of reward maximization; then, the Lagrange multiplier governing the constraint is updated using a heuristic rule. The approach is tested on various reasoning benchmarks, including AIME-24, MATH, and Olympiad, where the model performs as well as or better than its counterparts while producing more concise outputs.

**Strengths:**

- The proposed idea is notably simple and highly effective. Various problems in RL (for example, safety) involve multiple constraints. This paper effectively borrows from these approaches by formulating the length restriction as a constrained optimization problem.
- The proposed approach results in a minor modification to the standard training process for LLMs on reasoning tasks. Empirically, it achieves marginally better or similar performance compared to its baselines, while using fewer output tokens.
- The experiments are thorough, and the ablation studies are well-executed. The inclusion of OOD (out-of-distribution) results is a valuable addition, especially since Laconic does not suffer a significant drop in performance when evaluated on reasoning problems outside the training dataset's distribution.

**Weaknesses:**

- The primary weakness of the paper lies in the update rule for the Lagrange multiplier proposed in Eq. 6. It is unclear how that update rule is obtained, and it does not appear to follow the standard procedure of constrained optimization, where partial derivatives of the Lagrange function are set to zero and the system of equations is solved simultaneously. Furthermore, the cost expression in Eq. 3 is unbounded, and in certain cases, it might outweigh the reward term, resulting in undesirable updates. The approach prevents the LLM from producing responses beyond the specified token budget in expectation, but it does not necessarily result in succinct responses; for instance, “what is 2+3?” could still result in an unreasonably long response that is within the token budget, which is unnecessary.
- Another significant weakness is that the Lagrange multiplier and the token budget are prompt-agnostic. This may have an unwanted effect in cases where a longer response is required for some questions, especially when the questions are sampled from a heterogeneous mixture of datasets (also as noted above in the “what is 2+3?” example).
- On the experimental side, the paper would benefit from including results where the Lagrange multiplier is treated as a fixed hyperparameter and found via hyperparameter tuning. Such results would help to validate the benefits of the proposed update rule. The experiments section could be further strengthened by including more baselines, including certain prompt-based baselines that instruct the LLM to produce shorter reasoning traces and responses.

**Questions:**

- A surprising finding is the performance improvement on certain benchmarks when the response length is restricted, as the expectation would be to maintain the same performance or see a slight degradation. The authors should offer a hypothesis for this unexpected result.
- The paper should also address the counterintuitive performance drop observed across all datasets when the response length is increased from 2000 to 3000 tokens. This contradicts the improvement seen from 1500 to 2000 tokens and the general intuition that a larger budget should lead to better performance. An explanation for this anomaly is needed.
- The justification for the ceiling on the Lagrange multiplier is unclear. It appears arbitrary and lacks an obvious mathematical derivation. The authors should explain how this was determined.
- The paper would be strengthened by a discussion on selecting an appropriate token budget, particularly for unknown tasks. The authors should elaborate on the observed effects when the budget is set either too low or too high.

---

> ### Author Response · Authors · 2025-11-23
> **Authors' Responses (Part 1)**
>
> We thank the reviewer for the effort to review our work and for providing constructive feedback and insightful comments for our work. We respond to the reviewer's concerns as follows.
>
> > Comment 1: It is unclear how the update rule in equation (6) is obtained, and it does not appear to follow the standard procedure of constrained optimization.
>
> We thank the reviewer for this great question and we would like to clarify that equation (6) is exactly the standard dual update for our constrained objective. The constrained optimization problem of length-aware LLMs is formulated as
> $$\max_\theta E_{q\sim P(Q), o\sim\pi_\theta(\cdot|q)}[r(q,o)], \\; \textrm{s.t. }\\; E_{q\sim P(Q), o\sim\pi_\theta(\cdot|q)}[L(o)] \le B.$$
> The corresponding Lagrangian is
> $$\mathcal{L}(\theta,\lambda) = E_{q\sim P(Q), o\sim\pi_\theta(\cdot|q)}[r(q,o)] - \lambda\\, (\frac{E_{q\sim P(Q), o\sim\pi_\theta(\cdot|q)}[L(o)]}{B}-1), \quad \lambda\ge 0,$$
> and the problem equivalently translates to $\min_{\lambda\ge 0}\max_\theta \mathcal{L}(\theta,\lambda)$. Indeed, the standard primal-dual approach solves the primal variable $\theta$ and dual variable $\lambda$ iteratively with partial derivatives. Specifically, this gives us the gradient-descent-style dual update rule:
> $$\lambda \leftarrow \max\\{\lambda - \eta \frac{\partial \mathcal{L}(\theta,\lambda)}{\partial \lambda},0\\} = \max\\{\lambda + \eta (\frac{E_{q\sim P(Q), o\sim\pi_\theta(\cdot|q)}[L(o)]}{B}-1),0\\}.$$
> Empirically we estimate the expectation $E_{q\sim P(Q), o\sim\pi_\theta(\cdot|q)}[L(o)]$ with the minibatch mean $\bar{L}$, and we derive exactly equation (6):
> $$\lambda \leftarrow \max\\{\lambda + \eta(\frac{\bar{L}}{B} - 1), 0\\}.$$
> Therefore, equation (6) is the textbook projected gradient update for the Lagrange multiplier, applied with a Monte-Carlo estimate of the constraint, rather than an ad-hoc rule.
>
> However, using the purely linear cost function $\tilde{c}(q,o) = \frac{L(o) - B}{B}$ inside the primal (actor model) update is problematic in practice. Whenever $\lambda>0$, the actor model is incentivized to shorten responses on all samples, and collapses the policy to extremely short outputs. This leads to highly unstable training dynamics, which is undesirable in practice. We include the ablation experiments on the cost design in Appendix A of the revised version of our paper (c.f Table 5 and Figure 6 in the paper). The experimental results show that with linear cost function $\frac{L(o)-B}{B}$, the average accuracy plummets from 40\% to 8\%, and the mean response lengths fall below 10 tokens, indicating that the model is no longer producing meaningful responses during training.
>
> To improve the instability of model updates, we propose our clipped ReLU-style cost in the actor loss $c(q,o) = \max\\{\frac{L(o) - B}{B}, 0\\}$. As discussed in Section 2.2, this design preserves exploration as the model can freely use up to $B$ tokens without penalty (c.f line 149-150).
>
> ---
>
> > Comment 2: The cost expression in Eq. 3 is unbounded, and might outweigh the reward term, resulting in undesirable updates.
>
> As discussed in Section 4.3 (c.f. line 367-375), introducing a ceiling on $\lambda$ can avoid the cost outweighing the reward, and thus ensuring the preference of correct answers despite the lengths. The ablation experiments in Section 4.2 show that our method is robust with a $\lambda$-ceiling, and delivers virtually the same fine-tuned models.
>
> ---
>
> > Comment 3: The approach does not necessarily result in succinct responses; for instance, “what is 2+3?” could still result in an unreasonably long response that is within the token budget, which is unnecessary.
>
> Our objective in this work is to control expected response length under a global token budget. This corresponds to the practical setting where one wants to manage the overall compute cost of a deployed system (e.g., budget of a commercial API, or computing resource limits of an academic cluster), rather than enforce a per-prompt shortest possible answer. We recognize that determining the minimal sufficient length of a correct and clear response for each individual prompt (e.g., deciding exactly how terse "what is 2+3?" should be) is a challenging and independent problem.
>
> We agree that our approach does not guarantee that every single prompt receives the shortest conceivable response, and that some responses to easy prompts may remain somewhat verbose. We view prompt-aware budgets or prompt-dependent multipliers as natural extensions of our constrained RL formulation, and will add a discussion as promising future work. Our current contribution is to show that a simple and prompt-agnostic constrained RL formulation can yield a disciplined and effective mechanism for controlling average token usage.

---

> ### Author Response · Authors · 2025-11-23
> **Authors' Response (Part 2)**
>
> > Comment 4: The Lagrange multiplier and the token budget are prompt-agnostic. This may have an unwanted effect in cases where a longer response is required for some questions, especially when the questions are sampled from a heterogeneous mixture of datasets.
>
> In addition to our remarks on our prompt-agnostic formulation in the above response, we would like to add that a universal token budget constraint on the average response length allows the model to adaptively allocate tokens across a heterogeneous mixture of questions. Harder questions can still receive longer chains of thought, while easier ones are shortened, as long as the overall expected length remains under the budget. As observed empirically, our method reduces average length substantially while maintaining and sometimes improving accuracy across benchmark datasets.
>
> ---
>
> > Comment 5: The paper would benefit from including results where the Lagrange multiplier is treated as a fixed hyperparameter and found via hyperparameter tuning. Such results would help to validate the benefits of the proposed update rule.
>
> We thank the reviewer for the constructive suggestions. We agree that our paper would benefit from additional ablation experiments with fixed dual variables, and we add here the ablation experiments on fixed hyperparameters. We set the threshold $B=1000$ during training in all runs.
>
> **Table 1: Ablation experiments on fixed λ.**
>
> | Model              | AIME2024 |          | MATH |          | Minerva |          | Olympiad |          | Macro Avg |          |
> |--------------------|----------|----------|------|----------|---------|----------|----------|----------|-----------|----------|
> |                    | Pass@1   | # Tokens | Pass@1 | # Tokens | Pass@1 | # Tokens | Pass@1 | # Tokens | Pass@1    | # Tokens |
> | DeepScaleR-1.5B                |                 |                    |             |                |                 |                   |                  |                    |                   |                      |
> | + GRPO                         | 29.58           | 6122               | 85.28       | 1767           | 29.64           | 2630              | 49.11            | 3418               | 48.40             | 3484                 |
> | + fixed λ = 0.001              | 27.92           | 5834               | 82.73       | 2089           | 29.64           | 2200              | 46.76            | 3663               | 46.76             | 3396                 |
> | + fixed λ = 0.01               | 26.25           | 4709               | 81.65       | 1927           | 28.90           | 1990              | 45.52            | 2963               | 45.58             | 2897                 |
> | + fixed λ = 0.05               | 21.66           | 2417               | 78.28       | 1018           | 24.92           | 1073              | 41.25            | 1602               | 41.53             | 1528                 |
> | **+ `LACONIC`**                | 28.12           | 2665               | 83.75       | 1049           | 29.00           | 1189              | 46.83            | 1669               | 46.93             | 1643                 |
>
>
> As shown by the ablation results, with fixed Lagrange multipliers, the pass@1 scores are lower than both GRPO baseline and LACONIC. For small $\lambda$'s, the response lengths are moderately compressed compared to GRPO baseline, but significantly higher than LACONIC. For large $\lambda$'s, response lengths are substantially shortened, while pass@1 scores drop drastically. When the Lagrange multiplier is fixed, the algorithm reduces to heurisic reward design with a fixed length penalty, similar to the previous length-control method L1 [1]. As discussed in Section 1, these methods suffer from suboptimality in task reward or insufficient length control.

---

> ### Author Response · Authors · 2025-11-23
> **Authors' Response (Part 3)**
>
> > Comment 6: The experiments section could be further strengthened by including more baselines, including certain prompt-based baselines that instruct the LLM to produce shorter reasoning traces and responses.
>
> We agree that more baselines would further strengthen our paper. We add one more baseline ThinkPrune [2], kindly suggested by reviewer 9Kz2. The authors of [2] fine-tuned on the base model DeepScaleR-1.5B-Preview with two strategies, and published the two variants on HuggingFace. We evaluated the HuggingFace checkpoints of the two variants on AIME2024, MATH, Minerva, and Olympiad Bench. During all evaluations, the maximum context lengths are set to be 32,768, and we use the average of 16 samples with top\_p = 0.95, temperature = 0.6.
>
> **Table 2: Updated main experiment results.**
>
> | Model                 | AIME2024 |          | MATH  |          | Minerva |          | Olympiad |          | Macro Avg |          |
> |-----------------------|----------|----------|-------|----------|---------|----------|----------|----------|-----------|----------|
> |                       | Pass@1   | #Tokens | Pass@1 | #Tokens | Pass@1 | #Tokens | Pass@1  | #Tokens | Pass@1    | #Tokens |
> | DeepScaleR-1.5B   | 38.75           | 8140              | 85.92       | 3019           | 27.62           | 4134             | 51.94            | 5410              | 51.06             | 5176               |
> | + GRPO                | 29.58    | 6122     | 85.28 | 1767     | 29.64  | 2630     | 49.11   | 3418     | 48.40     | 3484     |
> | + L1-Exact            | 22.02    | 4138     | 82.57 | 3734     | 28.74  | 3755     | 44.91   | 3987     | 44.56     | 3904     |
> | + L1-Max              | 25.12    | 2979     | 83.45 | 1912     | 28.35  | 1701     | 44.76   | 2334     | 45.42     | 2231     |
> | + ThinkPrune2k        | 30.42    | 4275     | 83.52 | 1854     | 29.87  | 1961     | 46.57   | 2805     | 47.59     | 2724     |
> | + ThinkPrune-Iter2k   | 35.00    | 4598     | 84.22 | 1802     | 30.85  | 1941     | 48.77   | 2935     | 49.71     | 2819     |
> | **+ `LACONIC`**       | 28.12    | 2665     | 83.75 | 1049     | 29.24  | 1189     | 47.07   | 1669     | 47.05     | 1643     |
>
> From the evaluation results, our method LACONIC achieves comparable pass@1 compared to ThinkPrune2k and ThinkPrune-Iter2k with 0.54\% and 2.66\% drop respectively, while LACONIC uses 40\% and 42\% less tokens. **Overall, LACONIC provides a markedly better accuracy–length trade-off than existing length-control tuning algorithms.**
>
> Additionally, we add experiments on the Qwen3-8B model and present the results in the following table:
>
> **Table 3: Experiment results on Qwen3-8B.**
>
> | Model                 | AIME2024 |          | MATH  |          | Minerva |          | Olympiad |          | Macro Avg |          |
> |-----------------------|----------|----------|-------|----------|---------|----------|----------|----------|-----------|----------|
> |                       | Pass@1   | #Tokens | Pass@1 | #Tokens | Pass@1 | #Tokens | Pass@1  | #Tokens | Pass@1    | #Tokens |
> | Qwen3-8B     |                 |                   |             |                |                 |                  |                  |                   |                   |                    |
> | + GRPO       | 37.50           | 2918              | 87.95       | 1532           | 40.14           | 1549             | 54.40            | 2167              | 55.00             | 2042               |
> | **+ `LACONIC`** | 30.42        | 1643              | 87.01       | 792            | 38.46           | 783              | 53.05            | 1166              | 52.23             | 1096               |
>
> In the LACONIC training experiment, we set the token budget $B=1100$. The results on the 8B model show that LACONIC achieves comparable pass@1 compared to the GRPO baseline with a 2.73\% drop, while significantly reduce the response lengths by 46\%. **This shows that LACONIC can effectively shorten response lengths while preserving reasoning abilities on large reasoning models.**

---

> ### Author Response · Authors · 2025-11-23
> **Authors' Responses (Part 4)**
>
> We answer the reviewer's questions below.
>
> > Question 1: What is the performance improved on certain benchmarks when the response length is restricted, as the expectation would be to maintain the same performance or see a slight degradation.
>
> We thank the reviewer for the great question. The pass@1 scores of LACONIC on most benchmarks are higher than GRPO baseline (c.f. Table 3 in the paper), while all response lengths are shorter. We provide our interpretations of this result as follows. In the GRPO baseline experiment, we train the base model on GRPO with maximum response length set to 4K, same as in LACONIC experiments. This is to satisfy the computing resource limit, and also to keep in line with previous works [1,2]. As the base model DeepScaleR-1.5B-Preview supports up to 32,768 tokens, training with the 4K length cap will cause the performance to drop. While LACONIC pushes the response lengths below the token budget and recovers the performance with shortened responses, many responses generated by GRPO reach the length cap and are thus clipped, resulting in insufficient training on these prompts.
>
> We include the direct evaluations of the base model DeepScaleR-1.5B-Preview without further training on the restrictive response length cap in the first row of Table 2 above. As expected, the base model which is trained on a 32K context window has the highest pass@1 on most benchmarks. We would like to remark that our method LACONIC is able to achieve comparable reasoning performance (4\% drop on average) with 68\% less tokens, showing the superiority of LACONIC to effectively shorten response lengths while preserving reasoning abilities.
>
> ---
>
> > Question 2: The paper should also address the counterintuitive performance drop observed across all datasets when the response length is increased from 2000 to 3000 tokens.
>
> In these experiments, we set the maximum response length to be 4096. The 3000 token budget is hardly restrictive, where standard GRPO can naturally satisfy the constraint, and with LACONIC, $\lambda$ drops to 0 in early stages (c.f. Figure 3 (b),(c) in the paper). Due to the lack of model updates toward shorter responses, many responses reach the 4096 length cap and are clipped, leading to insufficient training on these prompts.
>
> ---
>
> > Question 3: The justification for the ceiling on the Lagrange multiplier is unclear. It appears arbitrary and lacks an obvious mathematical derivation.
>
> As discussed in Section 4.2, the ceiling on the Lagrange multiplier should be adopted when the maximum response length cap is large enough so that costs can outweigh rewards. We also derived the upper bound $\lambda_{\max} = \frac{B}{L_{\max} - B}$, where $L_{\max}$ is the maximum response length cap. Note that this upper bound is the largest value for $\lambda$-ceilings, and any $\lambda$-ceiling larger than it will not be fully effective. For our experiment settings, due to the limited computing resource, the maximum response lengths are low and a ceiling on $\lambda$ is not required. We thus set a restrictive $\lambda$-ceiling ($\lambda_{\max} = 0.1$) based on the training dynamics of runs without effective $\lambda$-ceilings.
>
> ---
>
> > Question 4: The paper would be strengthened by a discussion on selecting an appropriate token budget, particularly for unknown tasks.
>
> We thank the reviewer for the constructive suggestions. While the actual deployment budget can serve as a natural rule for selecting an appropriate token budget $B$, a simple and effective practice is to first train the base model with standard RL-tuning algorithms (e.g. GRPO) for some steps (we find 30 steps with batch size = 128 suffices). Then a proper and effective token budget $B$ can be selected to be 50-70\% of the average response lengths.
>
> ---
> References:
>
> [1] Pranjal Aggarwal and Sean Welleck. L1: Controlling how long a reasoning model thinks with
> reinforcement learning, 2025. URL: https://arxiv.org/abs/2503.04697.
>
> [2] Hou et al. ThinkPrune: Pruning Long Chain-of-Thought of LLMs via Reinforcement Learning, 2025. URL: https://arxiv.org/abs/2504.01296.

---

> > ### Comment · Reviewer_9Fv6 · 2025-11-24
> >
> > I thank the authors for their detailed response. I have reviewed their rebuttal, and it addresses the weaknesses and questions I raised in my review. However, I have a couple of follow-up inquiries before finalizing my evaluation:
> >
> > > The 3000 token budget is hardly restrictive, where standard GRPO can naturally satisfy the constraint, and with LACONIC, $\lambda$ drops to 0 in early stages (c.f. Figure 3 (b),(c) in the paper).
> >
> > Could the authors clarify what is meant by "naturally satisfy"? Does this imply that GRPO responses rarely exceed 3000 tokens? If so, this appears to contradict your earlier statement that _many responses generated by GRPO reach the length cap and are thus clipped, resulting in insufficient training on these prompts._ I encourage the authors to resolve this apparent inconsistency. Additionally, the authors remark that LACONIC is an adaptive method; what prevents the method from recovering once $\lambda$ drops to 0 initially?
> >
> > Additional follow-up questions:
> >
> > * To what value does $\lambda$ converge in the experiments?
> > * The Iter2k checkpoint performs significantly better while using fewer tokens. How does the performance scale when the token count is reduced further?
> > * The choice of $\lambda_\max=0.1$ seems quite restrictive. Does this imply that the constrained objective implicitly prioritizes the constraint over the reward maximization component?

---

> ### Author Response · Authors · 2025-11-24
> **Authors' Response to Follow-Up Questions (Part 1)**
>
> We sincerely thank the reviewer for the prompt response. We answer the reviewer's questions below.
>
> > Question 5: Could the authors clarify what is meant by "naturally satisfy"? Does this imply that GRPO responses rarely exceed 3000 tokens? If so, this appears to contradict your earlier statement that many responses generated by GRPO reach the length cap and are thus clipped, resulting in insufficient training on these prompts. I encourage the authors to resolve this apparent inconsistency.
>
> We would like to clarify that standard GRPO would naturally satisfy a $B=3000$ token budget **on average** (c.f. Figure 3(b)), not that individual GRPO responses rarely exceed 3000 tokens. The token-budget constraint $B$ is imposed on the expected response length, and in practice enforced via the average response length over prompts, rather than as a per-prompt constraint. As observed in Figure 3(b), when trained on GPRO with a 4K maximum response length, the model's average response length indeed stabilizes just below 3000 tokens, even though many individual responses are still very long.
>
> To make this clearer, we now report in Appendix B.2 the clip ratio (c.f. Figure 7), which is the fraction of responses in a minibatch that hit the maximum response length cap (4096 tokens in our settings). The logged clip ratio shows that 35-40\% responses generated by GRPO are clipped, whereas LACONIC keeps the clip ratio below 10\% (c.f. Figure 7(a)). This is consistent with our interpretation of why LACONIC performs better than GRPO on many benchmarks despite the restrictive token budget constraints, that a large clip ratio leads to insufficient training on those prompts.
>
> We also remark that this effect is much weaker for Qwen2.5-Math-1.5B-Instruct, which already outputs short responses with an extremely low clip ratio (<2\%, c.f. Figure 7(b)). In this case, the GRPO baseline on Qwen2.5-Math-1.5B-Instruct performs better than LACONIC on most benchmarks (c.f. Table 1 and 4 in the paper). This observation is also consistent with our interpretation. These results show that LACONIC has a clear advantage when baseline training induces many clipped, overlong responses under a tight token budget.
>
> ---
>
> > Question 6: The authors remark that LACONIC is an adaptive method; what prevents the method from recovering once $\lambda$ drops to 0 initially?
>
> We thank the reviewer for this insightful question. We offer the following clarifications:
>
> (1) LACONIC primarily recovers reasoning performance once $\lambda$ reaches 0. When $\lambda=0$, the LACONIC step reduces to the exact GRPO update, where length-aware costs vanish and the model purely optimizes the GRPO objective. This is clearly reflected in the training dynamics: as soon as $\lambda$ drops to 0, both the accuracy reward and average response lengths rapidly increase and then stabilize (c.f. Figure 3(a)-(c) and Figure 4(a)-(c)).
>
> (2) For DeepScaleR-1.5B, LACONIC(3000) is effectively GRPO after a short transient. As discussed above, a 3000-token budget is barely restrictive, and thus LACONIC quickly reduces to GRPO within around 20 steps (c.f. Figure 3(c)). This yields insufficient model updates toward shorter responses, resulting in the relatively low performance of LACONIC(3000) compared to LACONIC(2000,1750,1500) (c.f. Table 3 in the paper).
>
> (3) **LACONIC remains an adaptive method.** The dual update rule for $\lambda$ is adaptive, and the experiment results clearly show the different phases where LACONIC first enforces length constraints and then adaptively switches to pure objective maximization (c.f. Figure 3 and 4). Finally, once training has stabilized, LACONIC continues to adapt. Whenever the model begins to produce longer responses, $\lambda$ is driven back to a positive value, which penalizes excessive length and pulls the policy toward shorter responses (cf. Fig. 3(c)–(d)).
>
> ---
>
> > Question 7: To what value does $\lambda$ converge in the experiments?
>
> In the experiments, $\lambda$ stabilizes at 0 (c.f. Figure 3(c) and figure 4(c)), and the average response lengths stabilizes just below the token budget.

---

> ### Author Response · Authors · 2025-11-24
> **Authors' Response to Follow-Up Questions (Part 2)**
>
> > Question 8: The Iter2k checkpoint performs significantly better while using fewer tokens. How does the performance scale when the token count is reduced further?
>
> To the best of our knowledge, the authors of ThinkPrune-Iter2k have not released any trained checkpoints for further reduced response lengths. To examine the effect of scaling up the allowable length, we re-evaluate our LACONIC(2000) checkpoint using a 32,768 maximum response length. We report the updated evaluation results in the following table for convenience:
>
> **Table 4: Updated main experiment results.**
>
> | Model                 | AIME2024 |          | MATH  |          | Minerva |          | Olympiad |          | Macro Avg |          |
> |-----------------------|----------|----------|-------|----------|---------|----------|----------|----------|-----------|----------|
> |                       | Pass@1   | #Tokens | Pass@1 | #Tokens | Pass@1 | #Tokens | Pass@1  | #Tokens | Pass@1    | #Tokens |
> | DeepScaleR-1.5B   | 38.75           | 8140              | 85.92       | 3019           | 27.62           | 4134             | 51.94            | 5410              | 51.06             | 5176               |
> | + GRPO                | 29.58    | 6122     | 85.28 | 1767     | 29.64  | 2630     | 49.11   | 3418     | 48.40     | 3484     |
> | + L1-Exact            | 22.02    | 4138     | 82.57 | 3734     | 28.74  | 3755     | 44.91   | 3987     | 44.56     | 3904     |
> | + L1-Max              | 25.12    | 2979     | 83.45 | 1912     | 28.35  | 1701     | 44.76   | 2334     | 45.42     | 2231     |
> | + ThinkPrune2k        | 30.42    | 4275     | 83.52 | 1854     | 29.87  | 1961     | 46.57   | 2805     | 47.59     | 2724     |
> | + ThinkPrune-Iter2k   | 35.00    | 4598     | 84.22 | 1802     | 30.85  | 1941     | 48.77   | 2935     | 49.71     | 2819     |
> | **+ `LACONIC(1000)`**       | 28.12    | 2665     | 83.75 | 1049     | 29.24  | 1189     | 47.07   | 1669     | 47.05     | 1643     |
> | **+ `LACONIC(2000)`**       | 34.79   | 3976     | 85.61 | 1556     | 31.76  | 1869     | 48.96   | 2448     | 50.28     | 2462     |
>
> The experiment results show that LACONIC achieves 0.57\% higher pass@1 than ThinkPrune-Iter2k, while using 13\% less tokens.
>
> ---
>
> > Question 9: The choice of $\lambda_{\max} = 0.1$ seems quite restrictive. Does this imply that the constrained objective implicitly prioritizes the constraint over the reward maximization component?
>
> We remark that our choice of $\lambda_{\max}$ is precisely to avoid prioritizing the constraint over reward. With a proper ceiling $\lambda_{\max}$ ($\le B/(L_{\max} - B)$), any correct response always receives a higher Lagrangian reward $r - \lambda\\, c$ than any incorrect response, for all lengths up to $L_{\max}$ and for all $\lambda \in [0, \lambda_{\max}]$. In other words, the constraint penalty can never overturn the preference for correctness. Throughout training, the advantage signal always prefers correct responses over incorrect ones, and the reward term remains dominant.
>
> Moreover, our ablation on the step size $\eta$ shows that varying the effective scale of $\lambda$ leads to virtually identical trained checkpoints (c.f. Section 4.2). This indicates that LACONIC is robust to the precise magnitude of $\lambda$, and that in practice length control is achieved in a simple and stable way primarily by adjusting the token budget $B$, rather than by aggressively weighting the constraint term.

---

> ### Author Response · Authors · 2025-12-03
> **Authors' Response (Part 5)**
>
> We additionally evaluate LACONIC on DeepSeek-R1-Distill-Qwen-1.5B. We fine-tune the base model with a token budget $B=1500$ for 500 RL steps. At evaluation time, we compare our checkpoint against the base model, ThinkPrune-Iter2k [1], and Efficient-Reasoning [2] on AIME2024, MATH, Minerva, and Olympiad. The maximum response length is set to 32,768 during evaluation. The results are reported in Table 5.
>
> Table 5: Results on DeepSeek-R1-1.5B.
>
> | Model                 | AIME2024 |          | MATH  |          | Minerva |          | Olympiad |          | Macro Avg |          |
> |-----------------------|----------|----------|-------|----------|---------|----------|----------|----------|-----------|----------|
> |                       | Pass@1   | #Tokens | Pass@1 | #Tokens | Pass@1 | #Tokens | Pass@1  | #Tokens | Pass@1    | #Tokens |
> | DeepSeek-R1-1.5B      | 30.00           | 15218             | 82.90       | 5340          | 29.13          | 6783             | 44.8            | 12417             | 46.71        | 9940           |
> | + Efficient-Reasoning | 22.08           | 8801              | 81.63       | 1909          | 27.57          | 2018             | 42.96           | 4799              | 43.56        | 4382           |
> | + ThinkPrune-Iter2k   | 23.33           | 5440              | 81.35       | 1762          | 27.35          | 1837             | 43.39           | 3021              | 43.85        | 3015           |
> | **+ `LACONIC`**         | 24.17       | 5140          | 81.88   | 1636      | 28.36      | 1638         | 44.11       | 3241          | 44.63    | 2914       |
>
> These results show that LACONIC consistently achieves higher pass@1 with fewer tokens than existing length-control methods [1,2] on DeepSeek-R1-Distill-Qwen-1.5B. Relative to the original base model, LACONIC incurs only a modest macro-average performance drop of 2.08 percentage points, while using approximately 71\% fewer tokens on average.
>
> ---
>
> References
>
> [1] Hou et al. ThinkPrune: Pruning Long Chain-of-Thought of LLMs via Reinforcement Learning, 2025. URL: https://arxiv.org/abs/2504.01296.
>
> [2] Daman Arora and Andrea Zanette. Training Language Models to Reason Efficiently, 2025. URL: https://arxiv.org/abs/2502.04463

---

### Author Response · Authors · 2025-12-03
**Final Comments by Authors (Summary for ACs)**

Dear Area Chair,

We sincerely thank you for your time and efforts in handling our submission. For your convenience, we summarize below the main strengths highlighted by the reviewers and the key revisions and new results added after the rebuttal.

---

### Strengths Highlighted by Reviewers

1. **Novelty**
   - `kveD`: “The proposed solution is interesting. The method elegantly reinterprets length control as a constrained optimization problem rather than heuristic reward shaping.”
   - `9Kz2` also emphasizes that our work is “technically novel”.

2. **Significance**
   - `9Kz2`: “The method is well motivated: The problem of inference time costs is important, and the paper tackles it with an appropriate solution.”

3. **Soundness and Effectiveness**
   - `9Fv6`: “The experiments are thorough, and the ablation studies are well-executed”, and “The proposed idea is notably simple and highly effective”.
   - `kveD` further notes that the primal–dual approach is conceptually sound.

4. **Clarity and Presentation**
   - `9Kz2`: “The paper is well written and presented.”
   - `kveD`: “The paper is easy to follow.”

---

### Key Revisions and New Results

1. **Large-Scale Model Evaluation (Re: `9Kz2` and `kveD`)**
   We added new experiments on **Qwen3-8B**, showing that LACONIC’s length–accuracy trade-offs and efficiency gains extend to larger models under the same training recipe and unchanged inference-time interface.

2. **Expanded Baseline Comparison (Re: `9Kz2` and `kveD`)**
   We included results against **two additional strong baselines** including new results of **one additional base model**, further demonstrating that LACONIC consistently achieves substantially shorter responses with better accuracy under matched training and evaluation conditions.

3. **Primal–Dual Update Derivation (Re: `9Fv6`)**
   We provided a rigorous derivation of the primal–dual update rules and added it to the appendix, making the connection to standard constrained optimization Lagrangian methods explicit and formally justified.

4. **Questions About Experimental Results (Re: `9Fv6` and `kveD`)**
   We clarified our interpretation of the empirical behaviors raised by the reviewers and added new training-dynamics plots (clip ratio) as evidence to support our interpretation.

---

Most reviewers recognize the novelty and practical relevance of LACONIC, and the main concerns focus on model scale and the breadth of baselines. With the new Qwen3-8B experiments and additional baseline comparisons, we believe the revised results further validate LACONIC’s ability to significantly reduce token usage while maintaining or improving accuracy across models and benchmarks.

We again thank you for your careful consideration of our work.

Best regards,
The Authors

---

### Meta-Review · Area_Chair_Q7Ra · 2026-01-04

**Summary:**

This paper proposes LACONIC, a constrained reinforcement learning framework designed to control Large Language Model (LLM) output length by enforcing a token budget via a primal-dual approach. Major concerns from the reviewers are

- Lack of formal convergence analysis for the proposed update rule
- Prompt-agnostic constraints failing to guarantee succinctness on simple tasks
- Limited evaluation scope regarding RL algorithms (focused on GRPO)
- Initial lack of large-scale model validation
- Missing comparisons to recent length-control baselines

Meanwhile, the reviewers also acknowledged the contributions of this study, such as

- Novel formulation of length control as a constrained optimization problem
- Simple and empirically effective implementation
- Strong performance preservation on O.O.D. benchmarks

Overall, I believe this paper explores a practical and well-motivated problem in efficient LLM inference. While the rebuttal added larger model experiments and baselines, fundamental issues regarding the theoretical soundness and the granularity of the length constraint remain. Therefore, I encourage the authors to provide rigorous theoretical grounding and a more flexible constraint mechanism so that the work is more solid for publication on ICLR.

**Reviewer Concerns:**

The authors have answered most of the questions by the three reviewers (9Fv6, kveD, 9Kz2). However, several key concerns still remain:

- Lack of formal convergence analysis (kveD): The authors acknowledged the theoretical convergence proof as future work.

- Prompt-agnostic constraints / lack of instance succinctness (9Fv6)

- Unverified algorithmic generalizability (9Kz2): The authors conceded that extending validation to other algorithms like PPO is future work, leaving the claim of general flexibility unsupported.

**Reviewer Scores:**

I would expect most of the reviewer maintain their scores since their concerns are not fully addressed.

---

### Decision · Program_Chairs · 2026-01-26

Reject